# BOND: Benchmarking Unsupervised Outlier Node Detection on Static Attributed Graphs

**Kay Liu**[1,*], **Yingtong Dou**[1,8,*], **Yue Zhao**[2,*], **Xueying Ding**[2], **Xiyang Hu**[2],
**Ruitong Zhang**[3], **Kaize Ding**[4], **Canyu Chen**[5], **Hao Peng**[3], **Kai Shu**[5],
**Lichao Sun**[6], **Jundong Li**[7], **George H. Chen**[2], **Zhihao Jia**[2], **Philip S. Yu**[1]
[1]University of Illinois Chicago [2]Carnegie Mellon University
[3]Beihang University [4]Arizona State University [5]Illinois Institute of Technology
[6]Lehigh University [7]University of Virginia [8]Visa Research
`benchmark@pygod.org`

## Abstract

Detecting which nodes in graphs are outliers is a relatively new machine learning task with numerous applications. Despite the proliferation of algorithms developed in recent years for this task, there has been no standard comprehensive setting for performance evaluation. Consequently, it has been difficult to understand which methods work well and when under a broad range of settings. To bridge this gap, we present—to the best of our knowledge—the first comprehensive benchmark for unsupervised outlier node detection on static attributed graphs called BOND, with the following highlights. (1) We benchmark the outlier detection performance of 14 methods ranging from classical matrix factorization to the latest graph neural networks. (2) Using nine real datasets, our benchmark assesses how the different detection methods respond to two major types of synthetic outliers and separately to "organic" (real non-synthetic) outliers. (3) Using an existing random graph generation technique, we produce a family of synthetically generated datasets of different graph sizes that enable us to compare the running time and memory usage of the different outlier detection algorithms. Based on our experimental results, we discuss the pros and cons of existing graph outlier detection algorithms, and we highlight opportunities for future research. Importantly, our code is freely available and meant to be easily extendable:
`https://github.com/pygod-team/pygod/tree/main/benchmark`

## 1 Introduction

Outlier detection (OD) on a graph refers to the task of identifying which nodes in the graph are outliers. This is a key machine learning (ML) problem that arises in many applications, such as social network spammer detection [77], sensor fault detection [27], financial fraudster identification [22], and defense against graph adversarial attacks [33]. Unlike classical OD on tabular and time-series data, graph OD has additional challenges: (1) the graph data structure in general carries richer information, and thus more powerful ML models are needed to learn informative representations, and (2) with more complex ML models, training can be more computationally expensive in terms of both running time and memory consumption [35, 47], posing challenges for time-critical (i.e., low time budget) and resource-sensitive (e.g., limited GPU memory) applications.

Despite the importance of graph OD and many algorithms being developed for it in recent years, *there is no comprehensive benchmark on graph outlier detection*, which we believe has hindered the development and understanding of graph OD algorithms. In fact, a recent graph OD survey calls for "system benchmarking" and describes it as "the key to evaluating the performance of graph OD techniques" [55]. We remark that there already are benchmarks for general graph mining (e.g.,

---

*Equal Contribution.

OGB [30]), graph representation learning [26], graph robustness evaluation [95], graph contrastive learning [97], graph-level anomaly detection [85],[1] as well as benchmarks for tabular OD [6] and time-series OD [46]. These do not cover the specific task we consider, which we now formally define:

**Definition 1** *(Unsupervised Outlier Node Detection on Static Attributed Graphs (abbreviated as OND)) A static attributed graph is defined as $G = (V, E, \mathbf{X})$, where $V = \{1, 2, \ldots, N\}$ is the set of vertices, $E \subseteq \{(i, j) : i, j \in V \text{ s.t. } i \neq j\}$ is the set of edges, and $\mathbf{X} \in \mathbb{R}^{N \times D}$ is the node attribute matrix (the $i$-th row of $\mathbf{X}$ is the feature vector in $\mathbb{R}^D$ corresponding to the $i$-th node in the graph). Given the graph $G$, the goal of the problem OND is to learn a function $f : V \to \mathbb{R}$ that assigns a real-valued outlier score to every node in $G$. The outlier nodes are then taken to be the $k$ nodes with the highest outlier scores, for a user-specified value of $k$. This problem is unsupervised since in learning $f$, we do not have any ground truth information as to which nodes are outliers or not.*

Note that there are other graph OD problems (e.g., feature vectors could be time-dependent, there could be supervision in terms of some outliers being labeled, the nodes and edges could change over time, etc) but we focus on the problem OND stated above as it is the most prevalent [16, 55]. For OND, the status quo for how algorithms are developed has the following limitations:

- **Lack of a comprehensive benchmark:** often, only a limited selection of OND algorithms is tested on only a few datasets, making it unclear to what extent the empirical results generalize to a wider range of settings. Here, we remark that this issue of generalization is exacerbated by the fact that across different applications, what constitutes an outlier can vary drastically and, at the same time, also be difficult to precisely define in a manner that domain experts agree upon.
- **Limited outlier types taken into account:** typically, only a few types of outliers are considered (e.g., specific kinds of synthetic outliers are injected into real datasets), making it difficult to understand how graph OD algorithms respond to a wider variety of outlier nodes, including ones that are "organic" (non-synthetic).
- **Limited analyses of computational efficiency in both time and space:** Existing work mainly focuses on detection accuracy, with limited analyses of running time and memory consumption.

To address all the above limitations, we establish *the first comprehensive benchmark for the problem of OND* that we call BOND (short for benchmarking unsupervised outlier node detection on static attributed graphs). To accommodate many algorithms, we specifically create an open-source Python library for Graph Outlier Detection (PyGOD)[2], which provides more than ten of the latest graph OD algorithms, all with unified APIs and optimizations. Meanwhile, PyGOD also includes multiple non-graph baselines, resulting in a total of 14 representative and diverse methods for OND. We remark that this library can readily be extended to include additional OD algorithms.

Our work has the following highlights:

1. **The first comprehensive node-level graph OD benchmark**. We examine 14 OD methods, including classical and deep ones, and compare their pros and cons on nine benchmark datasets.
2. **Consolidated taxonomy of outlier nodes.** We group existing notions of outlier nodes into two main types: structural and contextual outliers. Our results show that most methods fail to balance the OD performance of these two major outlier types.
3. **Systematic performance flaw found for existing deep graph OD methods.** Surprisingly, our experimental results in BOND reveal that most of the benchmarked deep graph OD methods have suboptimal OD performance on organic outliers.
4. **Evaluation of both detection quality and computational efficiency**. In addition to common *effectiveness* metrics (e.g., ROC-AUC), we also measure the running time and GPU memory consumption of different algorithms as their *efficiency* measures.
5. **Reproducible and accessible benchmark toolkit**. To foster accessibility and fair evaluation for future algorithms, we make our code for BOND freely available at: https://github.com/pygod-team/pygod/tree/main/benchmark

We briefly describe existing approaches for OND in §2. We provide an overview of BOND in §3, followed by detailed experimental results and analyses in §4. We summarize the paper and discuss future work in §5.

---

[1]Graph-level anomaly detection refers to when we have a set of graphs and want to find which graphs are significantly different from the majority of graphs; in contrast, the graph OD we focus on in this paper is for detecting outliers at the node level for a specific graph.

[2]A Python library for Graph Outlier Detection (PyGOD): https://pygod.org/

## 2 Related Work

In this section, we briefly introduce related work on outlier node detection. Please refer to [2] and [55] for more comprehensive reviews of classical and deep-learning-based graph outlier detectors. We have implemented most of the discussed algorithms in this section in the PyGOD.

**Classical (non-deep) outlier node detection.** Real-world evidence suggests that the outlier nodes are different from regular nodes in terms of structure or attributes. Thus, early work on node outlier detection employs graph-based features such as centrality measures and clustering coefficients to extract the anomalous signals from graphs [2]. Instead of handcrafting features, learning-based methods have been used to more flexibly encode graph information to spot outlier nodes. Examples of these learning-based methods include ones based on matrix factorization (MF) [3, 50, 60, 69], density-based clustering [7, 29, 75], and relational learning [42, 63]. As most of the methods above have constraints on graph/node types or prior knowledge, we only include SCAN [75], Radar [50] and ANOMALOUS [60] in BOND to represent methods in this category.

**Deep outlier node detection.** The rapid development of deep learning and its use with graph data has shifted the landscape of outlier node detection from traditional methods to neural network approaches [55]. For example, the autoencoder (AE) [38], which is a neural network architecture devised to learn an encoding of the original data by trying to reconstruct the original data from the encoding, has become a popular model in detecting outlier nodes [16, 25, 39, 64, 70, 80]. AEs can be learned in an unsupervised manner as we are aiming to reconstruct the original data without separately trying to predict labels. The heuristic behind AE-based outlier detection is that we can use the AE reconstruction error as an outlier score; a data point that has a higher reconstruction error is likely more atypical.

More recently, graph neural networks (GNNs) have attained superior performance in many graph mining tasks [17, 18, 40, 71, 82]. GNNs aim to learn an encoding representation for every node in the graph, taking into account node attributes and also the underlying graph structure. The encoding representations learned by GNNs turn out to capture complex patterns that are useful for OD. As a result, GNNs have also become popular in detecting outlier nodes in graphs [19, 72, 87, 54, 76]. Note that GNNs can be combined with AEs; in constructing an AE, we need to specify encoder and decoder networks, which could be set to be GNNs.

We point out that it is also possible to adopt a Generative Adversarial Network (GAN) for outlier node detection [11]. GANs learn how to generate fake data that resemble real data by simultaneously learning a generator network (that can be used to randomly generate fake data) and a discriminator network (that tries to tell whether a data point is real or fake). Naturally, outliers could be deemed to be data points that are considered more "fake".

Among the many deep outlier node detectors, we have thus far implemented nine (see Table 2) for inclusion in BOND, where we have tried to have these be somewhat diverse in their methodology.

## 3 BOND

In this section, we provide an overview of BOND. We begin by defining two outlier types in §3.1. We then elaborate on the datasets (§3.2), algorithms (§3.3), and evaluation metrics (§3.4) in BOND.

### 3.1 Outlier Types

Many researchers have defined fine-grained outlier node types from different perspectives [2, 3, 16, 33, 50, 55]. In this paper, we group existing outlier node definitions into two major types according to real-world outlier patterns: *structural outliers* and *contextual outliers*, which are illustrated in Figure 1 and defined below.

**Definition 2** *(Structural outlier) Structural outliers are densely connected nodes in contrast to sparsely connected regular nodes.*

Structural outliers arise in many real-world applications. For example, members of organized fraud gangs who frequently collude in carrying out malicious activities can be viewed as forming dense subgraphs of an overall graph (with nodes representing different people) [2]. As another example, coordinated bot accounts retweeting the same tweet will form a densely-connected co-retweet graph [29, 58]. Note that some papers [50, 55] also regard isolated nodes that do not belong to any communities as structural outliers (i.e., they have only a few edges connecting to any communities), which is different from our definition above. Since there is no existing OD method that we are aware of for detecting these isolated outlier nodes, we do not cover this type of outlier in BOND.

**Definition 3** *(Contextual outlier) Contextual outliers are nodes whose attributes are significantly different from their neighboring nodes.*

An example of a contextual outlier is a compromised device in a computer network [2]. The definition of a contextual outlier is similar to how outliers are defined in classical proximity-based OD methods [1].

Some researchers call a node whose attributes differ from those of all other nodes as a *global outlier* [55] or a contextual outlier [50]. We do not consider these outliers in BOND as we find that they do not actually use the graph structure; instead, these outliers could be detected using tabular outlier detectors [28, 92].

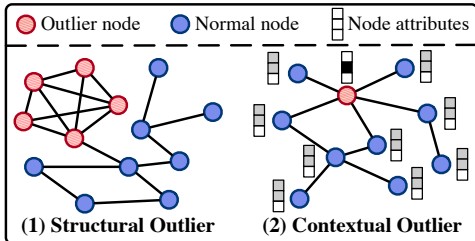

Figure 1: An illustration of structural vs. contextual outliers.

What we defined as a contextual outlier in Definition 3 is also referred to as an attribute outlier [16] or a community outlier [50, 55] in previous work. We argue that calling these *contextual outliers* is a more accurate terminology. The reason is that an "attribute outlier" sounds like it only depends on attributes (i.e., feature vectors), which would correspond to a global outlier [55], but confusingly this is not what is meant by the terminology. Meanwhile, the terminology of a "community" in graph theory has often been in reference to the density of edges among nodes and so a "community outlier" might be misconstrued to be what we call a *structural outlier* as in Definition 2. For the remainder of the paper, by "structural" and "contextual" outliers, we always go by Definitions 2 and 3 respectively.

Importantly, note that in real datasets, the organic (non-synthetic) outlier nodes present do not need to strictly be either a structural or a contextual outlier. In fact, what precisely makes them an outlier need not be explicitly stated and they could be neither a structural nor a contextual outlier, or they could even appear as a mixture of these two types! This makes detecting organic outliers more difficult than detecting synthetic outliers that follow a specific pattern such as those of Definitions 2 and 3.

### 3.2 Datasets

To comprehensively evaluate the performance of existing OND algorithms, we have investigated various real datasets with organic outliers used in previous literature. Note that some standard datasets are beyond the scope of the problem OND that we consider or do not make use of either the graph structure or node attributes/feature vectors. For example, YelpChi-Fraud [22], Amazon-Fraud [22], and Elliptic [74] are three graph datasets designed for supervised node classification; however, the fraudulent nodes have limited outlier pattern in terms of graph structure. Bitcoin-OTC, Bitcoin-Alpha, Epinions, and Amazon-Malicious from [43] are four bipartite graphs where nodes do not have attributes. DARPA [78], UCI Message [94], and Digg [94] are three dynamic graphs with organic edge outliers which are also out of our problem scope.

In BOND, we use the following datasets. First, since there are a limited number of open-source graph datasets with organic outlier nodes, we include three real datasets with no organic outliers that we inject synthetic outlier nodes into. Specifically, we use node classification benchmark datasets (**Cora** [66], **Amazon** [67], and **Flickr** [81]) from three domains with different scales. Next, we use six real datasets that contain organic outliers (**Weibo** [86], **Reddit** [44, 73], **Disney** [65], **Books** [65], **Enron** [65], and **DGraph** [32]). Finally, we also use purely synthetic data generated using the random algorithm by [36] that is able to produce graphs with varying scales; this random generation procedure provides a controlled manner in which we can evaluate different OD algorithms' computational efficiency in terms of both running time and memory usage. Some basic statistics for the real datasets used are given in Table 1 with more dataset details available in Appx. A.1.

To make synthetic outlier nodes of the two types we defined in §3.1 and to "camouflage" them so that they are more difficult to detect using simple OD methods, we slightly modify a widely-used approach [16, 25, 11, 80] (described below). These synthetic outliers are used with the real datasets that lack organic outliers (Cora, Amazon, Flickr) and also with the randomly generated graph data. In the random outlier injection procedures to follow, for the given graph $G$ that we are working with, we treat the vertex set as fixed. To inject structural outliers, we modify the edges present, whereas to inject contextual outliers, we modify the feature vectors of randomly chosen nodes.

**Injecting random structural outliers.** The basic strategy is to create $n$ non-overlapping densely connected groups of nodes, where each group has exactly $m$ nodes (so that there are a total of $m \times n$

Table 1: Statistics of real datasets used in BOND (* indicates that outliers are synthetically injected).

| Dataset | #Nodes | #Edges | #Feat. | Degree | #Con. | #Strct. | #Outliers | Ratio |
|---|---|---|---|---|---|---|---|---|
| **Cora**[*] [66] | 2,708 | 11,060 | 1,433 | 4.1 | 70 | 70 | 138 | 5.1% |
| **Amazon**[*] [67] | 13,752 | 515,042 | 767 | 37.2 | 350 | 350 | 694 | 5.0% |
| **Flickr**[*] [81] | 89,250 | 933,804 | 500 | 10.5 | 2,240 | 2,240 | 4,414 | 4.9% |
| **Weibo** [86] | 8,405 | 407,963 | 400 | 48.5 | - | - | 868 | 10.3% |
| **Reddit** [44, 73] | 10,984 | 168,016 | 64 | 15.3 | - | - | 366 | 3.3% |
| **Disney** [65] | 124 | 335 | 28 | 2.7 | - | - | 6 | 4.8% |
| **Books** [65] | 1,418 | 3,695 | 21 | 2.6 | - | - | 28 | 2.0% |
| **Enron** [65] | 13,533 | 176,987 | 18 | 13.1 | - | - | 5 | 0.4‰ |
| **DGraph** [32] | 3,700,550 | 4,300,999 | 17 | 1.2 | - | - | 15,509 | 0.4% |

Table 2: Algorithms implemented in BOND and their characteristics: whether designed for graphs (row 3), whether neural networks are used (row 4), and what the core idea for the method (row 5).

| Alg. | LOF | IF | MLPAE | SCAN | Radar | ANOMA-LOUS | GCNAE | DOMI-NANT | DONE/AdONE | Anomaly-DAE | GAAN | GUIDE | CONAD |
|---|---|---|---|---|---|---|---|---|---|---|---|---|---|
| Year | 2000 | 2012 | 2014 | 2007 | 2017 | 2018 | 2016 | 2019 | 2020 | 2020 | 2020 | 2021 | 2022 |
| Graph | ✗ | ✗ | ✗ | ✓ | ✓ | ✓ | ✓ | ✓ | ✓ | ✓ | ✓ | ✓ | ✓ |
| Deep | ✗ | ✗ | ✓ | ✗ | ✗ | ✗ | ✓ | ✓ | ✓ | ✓ | ✓ | ✓ | ✓ |
| Core | N/A | Tree | MLP+AE | Cluster | MF | MF | GNN+AE | GNN+AE | MLP+AE | GNN+AE | GAN | GNN+AE | GNN+AE |
| Ref. | [5] | [52] | [64] | [75] | [50] | [60] | [39] | [16] | [4] | [25] | [11] | [80] | [76] |

structural outliers injected). To do this, for each $i = 1, \ldots, n$, we randomly sample $m$ nodes to form the $i$-th group (these $m$ nodes are sampled uniformly at random from nodes that have not been previously chosen to form a group); for these $m$ nodes, we first make them fully connected and then drop each edge independently with probability $p$.

**Injecting random contextual outliers.** To inject a total of $o$ contextual outliers, we first sample $o$ nodes from the vertex set $V$ without replacement; these are the nodes whose attributes we aim to modify as to turn them into contextual outliers. We denote the set of these $o$ nodes as $V_c$ (so that $o = |V_c|$), and refer to the remaining nodes $V_r := V \setminus V_c$ as the "reference" set. For each node $i \in V_c$, we randomly choose $q$ nodes without replacement uniformly at random from the reference set $V_r$. Among these $q$ reference nodes chosen, we find the one whose attributes deviate the most (in terms of Euclidean distance) from those of node $i$. We then change the attributes of node $i$ to be the same as those of this most dissimilar reference node found.

For more details about synthetic outlier injection, see Appx A.1.3.

### 3.3 Algorithms

Table 2 lists the 14 algorithms evaluated in the benchmark and their properties. Our principle for selecting algorithms to implement in BOND is to cover representative methods in terms of the published time ("Year"), whether they use the graph structure ("Graph"), whether they use neural networks ("Deep"), and what the core idea is behind the method ("Core"). By including non-graph OD algorithms (LOF, IF, MLPAE), we can investigate the advantages and deficiencies of graph-based vs. non-graph-based OD algorithms in detecting outlier nodes. Similarly, incorporating three classical OD methods (clustering-based SCAN, MF-based Radar, and ANOMALOUS) helps us understand the performance of classical vs. deep OD methods. We select a wide array of GNN-based methods including the vanilla GCNAE; the classic DOMINANT; AnomalyDAE, an improved version of DOMINANT; and also GUIDE and CONAD, two state-of-the-art methods in this category with different data augmentation techniques. Besides GNN-based BOND methods, two methods encoding graph information using other models (DONE/AdONE and GAAN) are also included. Please refer to Appx. A.2 for a more detailed introduction of the methods benchmarked in BOND.

### 3.4 Evaluation Metrics

**Detection quality measures**. We follow the extensive literature in graph OD [15, 69, 87] to comprehensively evaluate the outlier node detection quality with three metrics: (1) ROC-AUC reflects detectors' performance on both positive and negative examples, while (2) Average Precision focuses more on positive examples, and (3) Recall@k evaluates the examples with high predicted outlier scores. See Appx. A.3 for more details.

**Efficiency measures in time and space**. Another important aspect of graph-based algorithms is their high time and space complexity [20, 35], which imposes additional challenges for large, high-dimensional datasets on hardware like GPUs with limited memory (e.g., out-of-memory errors).

Table 3: ROC-AUC (%) comparison among OD algorithms on three datasets with synthetic outliers, where we show *the avg perf.* ± *the STD of perf.* (*max perf.*) of each. The best algorithm by expectation is shown in **bold**, while the max performance per dataset is marked with underline. OOM denotes out of memory with regard to GPU (_G) and CPU (_C).

| Algorithm | Cora | Amazon | Flickr |
|---|---|---|---|
| LOF | 69.9±0.0 (69.9) | 55.2±0.0 (55.2) | 41.6±0.0 (41.6) |
| IF | 64.4±1.5 (67.4) | 51.3±3.0 (57.9) | 57.1±1.1 (58.8) |
| MLPAE | 70.9±0.0 (70.9) | 74.2±0.0 (74.2) | 72.4±0.0 (72.5) |
| SCAN | 62.8±4.5 (72.6) | 62.2±4.9 (71.1) | 62.4±12.4 (75.0) |
| Radar | 65.0±1.3 (66.0) | 71.8±1.1 (73.4) | OOM_G |
| ANOMALOUS | 55.0±10.3 (68.0) | 72.5±1.5 (75.5) | OOM_G |
| GCNAE | 70.9±0.0 (70.9) | 74.2±0.0 (74.2) | 71.6±3.1 (72.4) |
| DOMINANT | 82.7±5.6 (84.3) | 81.3±1.0 (82.2) | 78.0±12.0 (84.6) |
| DONE | 82.4±5.6 (87.9) | 82.8±8.8 (93.7) | **84.7**±2.5 (89.0) |
| AdONE | 81.5±4.5 (87.4) | **86.6**±5.6 (92.3) | 82.8±3.2 (89.0) |
| AnomalyDAE | **83.4**±2.3 (85.3) | 85.7±2.9 (90.8) | 65.6±3.5 (70.4) |
| GAAN | 74.2±0.9 (76.1) | 80.8±0.3 (81.5) | 72.4±0.2 (72.5) |
| GUIDE | 74.7±1.3 (77.5) | OOM_C | OOM_C |
| CONAD | 78.8±9.6 (84.3) | 80.5±4.0 (82.2) | 65.1±2.5 (67.4) |

Table 4: ROC-AUC (%) comparison among OD algorithms on six datasets with organic outliers, where we show *the avg perf.* ± *the STD of perf.* (*max perf.*) of each. The best algorithm by expectation is shown in **bold**, while the max performance per dataset is marked with underline. OOM denotes out of memory with regard to GPU (_G) and CPU (_C). TLE denotes time limit of 24 hours exceeded.

| Algorithm | Weibo | Reddit | Disney | Books | Enron | DGraph |
|---|---|---|---|---|---|---|
| LOF | 56.5±0.0 (56.5) | **57.2**±0.0 (57.2) | 47.9±0.0 (47.9) | 36.5±0.0 (36.5) | 46.4±0.0 (46.4) | TLE |
| IF | 53.5±2.8 (57.5) | 45.2±1.7 (47.5) | **57.6**±2.9 (63.1) | 43.0±1.8 (47.5) | 40.1±1.4 (43.1) | **60.9**±0.7 (62.0) |
| MLPAE | 82.1±3.6 (86.1) | 50.6±0.0 (50.6) | 49.2±5.7 (64.1) | 42.5±5.6 (52.6) | 73.1±0.0 (73.1) | 37.0±1.9 (41.3) |
| SCAN | 63.7±5.6 (70.8) | 49.9±0.3 (50.0) | 50.5±4.0 (56.1) | 49.8±1.7 (52.4) | 52.8±3.4 (58.1) | TLE |
| Radar | **98.9**±0.1 (99.0) | 54.9±1.2 (56.9) | 51.8±0.0 (51.8) | 52.8±0.0 (52.8) | **80.8**±0.0 (80.8) | OOM_C |
| ANOMALOUS | **98.9**±0.1 (99.0) | 54.9±5.6 (60.4) | 51.8±0.0 (51.8) | 52.8±0.0 (52.8) | **80.8**±0.0 (80.8) | OOM_C |
| GCNAE | 90.8±1.2 (92.5) | 50.6±0.0 (50.6) | 42.2±7.9 (52.7) | 50.0±4.5 (57.9) | 66.6±7.8 (80.1) | 40.9±0.5 (42.2) |
| DOMINANT | 85.0±14.6 (92.5) | 56.0±0.2 (56.4) | 47.1±4.5 (54.9) | 50.1±5.0 (58.1) | 73.1±8.9 (85.0) | OOM_C |
| DONE | 85.3±4.1 (88.7) | 53.9±2.9 (59.7) | 41.7±6.2 (50.6) | 43.2±4.0 (52.6) | 46.7±6.1 (67.1) | OOM_C |
| AdONE | 84.6±2.2 (87.6) | 50.4±4.5 (58.1) | 48.8±5.1 (59.2) | 53.6±2.0 (56.1) | 44.5±2.9 (53.6) | OOM_C |
| AnomalyDAE | 91.5±1.2 (92.8) | 55.7±0.4 (56.3) | 48.8±2.2 (55.4) | **62.2**±8.1 (73.2) | 54.3±11.2 (69.1) | OOM_C |
| GAAN | 92.5±0.0 (92.5) | 55.4±0.4 (56.0) | 48.0±0.0 (48.0) | 54.9±5.0 (61.9) | 73.1±0.0 (73.1) | OOM_C |
| GUIDE | OOM_C | OOM_C | 38.8±8.9 (52.5) | 48.4±4.6 (63.5) | OOM_C | OOM_C |
| CONAD | 85.4±14.3 (92.7) | 56.1±0.1 (56.4) | 48.0±3.5 (53.1) | 52.2±6.9 (62.9) | 71.9±4.9 (84.9) | 34.7±1.2 (36.5) |

Therefore, we measure efficiency in time and space respectively, we use (1) wall-clock time and (2) GPU memory consumption. We provide more details in Appx. A.3.

## 4 Experiments

We design BOND to understand the detection effectiveness and efficiency of various OD algorithms in addressing the problem OND. Specifically, we aim to answer: **RQ1** (§4.1): How effective are the algorithms on detecting synthetic and organic outliers? **RQ2** (§4.2): How do algorithms perform under two types of synthetic outliers (structural and contextual)? **RQ3** (§4.3): How efficient are algorithms in terms of time and space? Note that due to space constraints, for detection quality, we focus on the ROC-AUC metric, deferring results using the AP and Recall@k metrics to Appx. C.

**Model implementation and environment configuration.** Most algorithms in BOND are implemented via our newly released PyGOD package [53], and non-graph OD methods are imported from our earlier work [92]. Although we tried our best to apply the same set of optimization techniques, e.g., vectorization, to all methods, we suspect that further code optimization is possible. For more implementation details and environment configurations, see Appx. B.

**Hyperparameter grid.** In real-world settings, it is unclear how to do hyperparameter tuning and algorithm selection for unsupervised outlier detection due to the lack of ground truth labels and/or universal criteria that correlates well with the ground truth [55, 93]. For fair evaluation, when we report performance metrics in tables, we apply the same hyperparameter grid (see Appx. B) to each applicable algorithm and report its *avg. performance* (i.e., "algorithm performance in expectation"), along with the *standard deviation* (i.e., "algorithm stability") and the *max* (i.e., "algorithm potential").

### 4.1 Experimental Results: Detection Performance on Synthetic and Organic Outliers

Using the nine real datasets described in §3.2, we report the ROC-AUC score of different OD algorithms in Tables 3 and 4. Below are the key findings from these tables.

**In terms of avg. performance, no outlier node detection method is universally the best on all datasets.** Tables 3 and 4 show that only three of 14 methods evaluated (AnomalyDAE, Radar, ANOMALOUS) have the best avg. performance (for the ROC-AUC metric) on two datasets (Cora and Books for AnomalyDAE; Weibo and Enron for Radar and ANOMALOUS). The classical methods Radar and ANOMALOUS both have the best performance on Weibo and Enron (see Table 4) but they are worse than many deep learning methods on detecting synthetic outliers (see Table 3). Additionally, there is a substantial performance gap between the best- and worst-performing algorithms, e.g., DONE achieves $2.06\times$ higher average ROC-AUC compared to LOF on Flickr.

**Most methods evaluated fail to detect organic outliers.** Since most methods we evaluated are designed to handle structural and contextual outliers as defined in §3.1, to figure out the reason behind the failure and success in detecting organic outliers, we analyze the organic outlier patterns in terms of metrics related to the definitions of structural and contextual outliers. We first show that the success of most methods on Weibo (see Table 4) is because the outliers in Weibo exhibit the properties of both structural and contextual outliers. Specifically, in Weibo, the average clustering coefficient [24] of the outliers is higher than that of inliers (0.400 vs. 0.301), meaning that these outliers correspond to structural outliers. Meanwhile, the average neighbor feature similarity [22] of the outliers is far lower than that of inliers (0.004 vs. 0.993), so that the outliers also correspond to contextual outliers. In contrast, the outliers in the Reddit and DGraph datasets have similar average neighbor feature similarities and clustering coefficients for outliers and inliers. Therefore, their abnormalities rely more on outlier annotations with domain knowledge, and so supervised OD methods are more effective than unsupervised ones on Reddit (best AUC: 0.746 in [73] vs. 0.604 in Table 4) and DGraph (best AUC: 0.792 in [32] vs. 0.620 in Table 4) than unsupervised ones.

**Deep learning methods and other methods using SGD may be sub-optimal on small graphs.** The outliers on Disney, Books, and Enron also have similar outlier patterns defined in §3.1. However, most of the deep learning methods evaluated do not work particularly well on Disney and Enron compared to classical baselines. The reason is that Disney and Books have small graphs in terms of #Nodes, #Edges, and #Feat. (see Table 1). The small amount of data could make it difficult for the deep learning methods to encode the inlier distribution well and could also possibly lead to overfitting issues. Meanwhile, classical methods Radar and ANOMALOUS also perform poorly on Disney and Books; these methods use SGD, which we suspect could be problematic for such small datasets.

**Different categories of methods evaluated are good at detecting different types of outliers.** According to Tables 3 and 4, many deep graph-based methods are good at detecting synthetic outliers but are useless in detecting organic outliers. Meanwhile, non-graph-based methods have advantages when outliers do not follow taxonomies (Reddit and DGraph). These observations corroborate our understanding of unsupervised OD algorithms—their effectiveness depends on whether the underlying data distribution satisfies structural properties that the algorithms exploit.

**In terms of standard deviation of the ROC-AUC metric, among the deep learning methods, some are noticeably less stable than others.**[1] Certain deep learning methods, e.g., GAAN, exhibit insensitivity to hyperparameters, where the ROC-AUC standard deviation is mostly below $1\%$. Meanwhile, the methods that are unstable tend to involve more complex loss terms (e.g., weighted combination of multiple losses); for instance, DONE and AdONE achieve the highest *max* performance (i.e., the numbers in parentheses in Tables 3 and 4, and not the *avg.* performance) among deep graph methods on three datasets, while showing high ROC-AUC standard deviation across hyperparameters tested. Here, we emphasize that in practice for unsupervised OD, there being no labels means that hyperparameter tuning is far less straightforward, so stability (in OD detection quality) across hyperparameters is a desirable property of an algorithm.

### 4.2 Experimental Results: Detection Performance on Structural and Contextual Outliers

We report the ROC-AUC metric of different algorithms on three datasets with two types of injected synthetic outliers (contextual and structural outliers from §3.2) in Table 5. Note that we only consider synthetic outliers here since we have generated them so that we know exactly which nodes are contextual vs. structural outliers. Our main findings are as follows.

---

[1] For a specific method, the standard deviation of the method's ROC-AUC scores across hyperparameters (what we have called "algorithm stability") and the maximum ("algorithm potential") are in general monotonically related (i.e., when the standard deviation increases, then the maximum *minus the mean* also tends to increase; and vice versa), so that our finding here is both for algorithm stability and potential.

Table 5: ROC-AUC (%) comparison among OD algorithms on three datasets injected with contextual and structural outliers, where we show *the avg perf.* ± *the STD of perf.* (*max perf.*) of each. The best algorithm by expectation is shown in **bold**, while the max performance per dataset is marked with underline. OOM denotes out of memory with regard to GPU (_G) and CPU (_C). Reconstruction-based MLPAE and GCNAE perform best w.r.t contextual outliers, while there is no universal winner for both types of outliers.

| Algorithm | Cora | | Amazon | | Flickr | |
|---|---|---|---|---|---|---|
| | Contextual | Structural | Contextual | Structural | Contextual | Structural |
| LOF | 87.1±0.0 (87.1) | 52.4±0.0 (52.4) | 61.9±0.0 (61.9) | 48.0±0.0 (48.0) | 34.2±0.0 (34.2) | 49.1±0.0 (49.1) |
| IF | 77.5±2.2 (81.8) | 51.4±2.3 (56.2) | 51.8±6.0 (64.3) | 50.9±0.8 (52.2) | 63.1±2.1 (66.5) | 50.9±0.3 (51.5) |
| MLPAE | **88.9**±0.0 (88.9) | 52.5±0.0 (52.5) | **98.6**±0.0 (98.6) | 49.0±0.0 (49.0) | **94.4**±0.1 (94.5) | 50.0±0.1 (50.3) |
| SCAN | 49.8±0.5 (51.7) | 80.0±13.4 (95.9) | 48.7±1.1 (49.9) | 78.0±11.9 (94.0) | 50.2±0.1 (50.3) | 86.8±21.1 (99.7) |
| Radar | 50.2±0.6 (51.0) | 78.4±3.4 (81.6) | 84.9±3.7 (88.2) | 59.0±1.7 (61.3) | OOM_G | OOM_G |
| ANOMALOUS | 51.1±1.3 (53.5) | 69.3±16.2 (90.8) | 85.4±0.9 (87.2) | 59.5±2.5 (62.7) | OOM_G | OOM_G |
| GCNAE | **88.9**±0.0 (88.9) | 52.5±0.0 (52.5) | **98.6**±0.0 (98.6) | 49.0±0.0 (49.0) | 88.7±9.2 (94.5) | 50.0±0.2 (50.3) |
| DOMINANT | 71.9±6.6 (74.4) | 93.0±4.6 (95.3) | 69.0±3.6 (71.3) | 93.6±2.8 (94.4) | 69.0±4.5 (71.0) | **96.3**±10.4 (98.9) |
| DONE | 70.2±8.3 (80.0) | 92.8±5.6 (99.8) | 82.4±11.1 (95.1) | 90.2±8.0 (99.4) | 85.7±2.1 (88.7) | 85.5±3.1 (89.1) |
| AdONE | 73.9±5.0 (78.0) | 91.3±4.8 (97.3) | 78.0±10.6 (95.1) | 85.7±11.8 (97.0) | 80.2±4.3 (88.1) | 87.1±1.9 (89.7) |
| AnomalyDAE | 80.2±2.8 (87.2) | 90.2±4.6 (95.9) | 88.2±9.6 (98.3) | 85.5±10.1 (94.3) | 80.0±7.4 (93.3) | 56.6±1.7 (59.0) |
| GAAN | 88.7±0.1 (88.8) | 61.0±0.8 (62.5) | 98.5±0.1 (98.6) | 64.2±2.0 (67.3) | 94.1±0.3 (94.4) | 50.3±0.3 (50.9) |
| GUIDE | 88.3±0.8 (88.7) | 61.8±2.4 (71.1) | OOM_C | OOM_C | OOM_C | OOM_C |
| CONAD | 72.5±5.8 (74.4) | **93.6**±4.8 (95.4) | 69.4±2.8 (71.3) | **94.1**±0.4 (94.3) | 65.8±0.9 (67.4) | 68.3±0.6 (69.1) |

**For GNNs, the reconstruction of the structural information appears to play a significant role in detecting structural outliers.** Specifically, the performance gap on structural outliers between GCNAE and DOMINANT is over 40%. By taking a closer look into these two algorithms, they differ only in that DOMINANT has a structural decoder that aims to reconstruct the adjacency matrix of the graph. That DOMINANT performs significantly better than GCNAE suggests that reconstructing information on graph structure is helpful in identifying structural outliers, which intuitively makes sense as these outliers are defined in terms of graph structure.

**Low-order structural information (i.e., one-hop neighbors) is sufficient for detecting structural outliers**. DOMINANT and DONE achieve similar mean ROC-AUC scores (~92%) on detecting structural outliers in the Cora and Amazon datasets even though DOMINANT encodes 4-hop neighbor information whereas DONE only encodes 1-hop neighbor information. This observation could facilitate outlier node detection model design since encoding high-order information usually imposes a higher computational cost, and multi-hop neighbor aggregation may even lead to the over-smoothing problem in GNNs [8].

**No method achieves high detection accuracy for both structural and contextual outliers.** For instance, none of the methods reaches 85% detection AUC on both structural and contextual outliers. Moreover, on Flickr with structural outliers injected, most attributed graph OD methods that are supposed to detect structural outliers have worse average ROC-AUC scores than that of SCAN, whereas SCAN is a non-attributed graph OD method detecting structural outliers by clustering on nodes. The above result suggests that the common approach that arbitrarily combines the structural and contextual loss terms with fixed weights (that are hyperparameters) can struggle to balance performance in detecting both outlier types. How to detect these two different types of outliers consistently well remains an open question.

We visualize the efficiency in time (wall clock running time) and space (GPU memory consumption) of selected algorithms in Figure 2. The complete results are available in Appx. C.3. All algorithms are evaluated under randomly generated graphs with the same sets of injected outliers to guarantee fairness. The running time in Figure 2 (left) is the sum of training and computing outlier scores. We only measure the GPU memory consumption of the different methods (as opposed to the CPU memory consumption) because it is often the bottleneck of OD algorithms [89]. For more details on the generated graphs and experimental settings of this section, see Appx. A.1.2 and Appx. B, respectively. Our key findings from Figure 2 are as follows.

### 4.3 Experimental Results: Computational Efficiency

**Time efficiency.** Classical methods that we evaluated take less time than the deep learning ones, which tend to be more complicated and learn more flexible models. Among the GNN-based methods, GUIDE consumes far more time compared to others. The reason is that GUIDE uses a graph motif counting algorithm (which is #P-complete [13]) to extract the structural features and consumes much more time on the CPU. CONAD takes the second-most amount of time due to its use of contrastive learning (which uses pairwise comparisons within mini-batches).

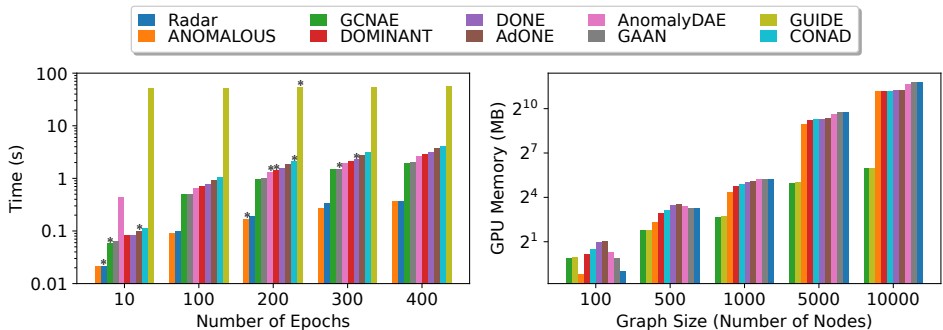

Figure 2: Wall-clock running time (left) and GPU memory consumption (right) of different methods. (∗ denotes the best performance of each method among five different numbers of epochs.)

**Space efficiency.** According to Figure 2 (right), GCNAE and GUIDE consume much less GPU memory than the other methods as the graph size increases. GCNAE saves more memory due to its simpler architecture. GUIDE consumes more CPU time and RAM to extract low-dimensional node motif degrees, thereby saving more GPU memory. Though classical methods have the advantage in terms of running time, most of them cannot be deployed in a distributed fashion due to the limitation of "global" operators like matrix factorization and inversion. One advantage of deep models is that they can be easily extended to minibatch and distributed training via graph sampling. Another advantage is that deep methods can be easily integrated with existing deep learning pipelines (e.g., graph pretraining module that obtains node embeddings).

## 5  Discussion

We have established BOND, the first comprehensive benchmark for unsupervised outlier node detection on static attributed graphs. Our benchmark has empirically examined the effectiveness of a diverse collection of OD algorithms in terms of synthetic vs. organic outliers, structural vs. contextual outliers, and computational efficiency. Importantly, a major goal in our development of BOND is to make it easy to extend so that further progress can be made in better understanding existing algorithms and developing new ones to address OND. We conclude this paper by discussing future research directions for OND in general (§5.1), and then specific to benchmarking (§5.2).

### 5.1  Future Directions in Addressing the Problem OND

Our experimental results on real data (§4.1 and §4.1) reveal substantial detection performance differences between algorithms, with none of them being the universal winner. Even for a single algorithm, there is also the issue of hyperparameter tuning (e.g., the detection performance of AnomalyDAE on Weibo varies by as much as 14% across hyperparameters). However, the fundamental problem is that because the problem OND is unsupervised, it is not straightforward deciding on the "right" choice of algorithm or hyperparameter setting. Any quantitative metric we define to help with model selection or hyperparameter tuning will require assumption(s). Separately, the available computational budget and the size of the dataset to be analyzed can limit what methods can even be used (our results in §4.3 show that some methods are much more computationally expensive than others).

*Opportunity 1: designing "type-aware" detection algorithms*. A major finding of our experimental results is that which OD algorithm works best heavily depends on the type of outlier encountered (synthetic vs. organic, structural vs. contextual). Put another way, if we expect to see a particular type of outlier, then this should inform the choice of OD algorithm to apply (e.g., MLPAE and GCNAE for contextual outliers). Of course, this would require us to have some a priori knowledge or guess as to what the outliers look like in a dataset. Importantly, in real applications, we might not need to detect *all* outliers. For example, practitioners may want to focus only on high-value or high-interest outliers (e.g., illegal trades that affect revenue the most [37] can be considered as contextual outliers, so MLPAE and GCNAE could be good choices). Following recent advances in categorizing outlier types in tabular data [34], we call for attention to identifying more fine-grain outlier types in OND and figuring out which algorithms are well-suited to these different outlier types. Once we have this sort of information, there could be opportunities for automatically choosing a single or combining multiple OD algorithms, accounting for outlier types (e.g., using an ensembling approach like [91]).

*Opportunity 2: synthesizing more realistic and flexible outlier nodes*. The organic outliers encountered in our real data experiments (§4.1) can be complex and composed of multiple outlier types

(possibly including types beyond just the structural and contextual ones we focused on). Our experimental results show that there is a wide detection performance gap on synthetic vs. organic outliers, calling for more realistic outlier generation approaches. To improve existing generation methods in BOND, we could use a generative model to fit the normal samples, and then perturb the generative model to generate different types of outliers. This approach has been successful in tabular OD [68].

***Opportunity 3: understanding the sensitivity of OND algorithms to hyperparameters and designing more stable methods***. We had pointed out in §4.1 that some deep learning methods are more stable (across hyperparameters) than others in terms of standard deviation in achieved ROC-AUC scores. This phenomenon extends of course to classical methods as well. Better understanding the drivers of these algorithms' (in)sensitivity to hyperparameters would help us better design algorithms that are more stable with respect to hyperparameter settings. In turn, this could help ease the burden of unsupervised hyperparameter tuning. In tabular OD tasks, researchers have developed methods for outlier detection that are more stable with respect to hyperparameters such as the robust autoencoder [96], RandNet [9], ROBOD [21], and ensemble frameworks [91]. Perhaps some techniques from these tabular OD methods could be incorporated into specific methods for OND to improve stability.

***Opportunity 4: developing more efficient OND algorithms.*** Our results on computational efficiency (§4.3) show that some algorithms take substantially more time and/or memory to execute than others. Meanwhile, most algorithms tested ran out of memory on the million-scale DGraph dataset (Table 4). We suggest developing more scalable algorithms for OND, which could mean more optimized implementations of existing algorithms and also the development of new algorithms. We point out several lines of work that could be helpful in this endeavor. First, there are existing approaches for making GNNs more scalable [35, 49] but these have yet to be specialized to address OND. Separately, most existing autoencoder-based methods that we tested reconstruct a complete graph adjacency matrix (e.g., DOMINANT, DONE, AnomalyDAE, GAAN), which is memory intensive (scaling quadratically with the number of nodes). Developing a more memory-efficient implementation of this step would be interesting. Next, approximating the node motif degree in GUIDE is possible, which can significantly reduce both computation time and space [10, 79]. Lastly, we mention that some recent work [90, 89] accelerates tabular OD via distributed learning, data and model compression, and/or quantization. These ideas could be extended to algorithms for OND.

***Opportunity 5: meta-learning to assist model selection and hyperparameter tuning***. Recent work on general graph learning [59] and unsupervised OD model selection on tabular data [93] shows that under meta-learning frameworks, we can identify good OD models to use for a new task (or dataset) based on its similarity to meta-tasks where ground truth information is available. A similar approach also works for unsupervised hyperparameter tuning [88]. We suggest exploring a meta-learning framework for algorithm selection and hyperparameter tuning in solving OND, including quantifying task similarity between graph OD datasets. Such a framework would require some but not all datasets to have ground truth, and that datasets with ground truth can be related to the ones without.

## 5.2  Future Directions in Improving Our Benchmark System BOND

**Extending detection tasks to different "levels".** In BOND, we focus on node-level detection with static attributed graphs due to their popularity, while there are more detection tasks at different levels of a graph. Recent graph OD algorithms extend to edge- [83], subgraph- [70], and graph-level [62, 85] detection. Future comprehensive graph OD benchmarks can include these emerging graph OD tasks.

**Incorporating supervision.** Although BOND focuses on unsupervised methods, there can be cases where a small set of labels (either for OD or relevant tasks) are available so that (semi-)supervised learning is possible (e.g., [22, 84]). Extending BOND to handle supervision would particularly be beneficial in addressing algorithm selection and hyperparameter tuning challenges.

**Curating more datasets.** Thus far, we have only included nine real datasets in BOND. Adding more datasets over time would be beneficial, especially ones with organic outliers. With a much larger collection of datasets (e.g., $\geq 20$), one could run statistical tests for comparison [14], which has been used in OD tasks with tabular datasets [51, 93]. Similar to tabular OD [23], one can convert existing multi-classification graph datasets (e.g., ones from Open Graph Benchmark (OGB) [30]) into OD datasets by treating one or combining several small classes to be treated as a single "outlier" class, with all other classes considered "normal".

## Acknowledgments

**Funding.** This work was supported in part by NSF under grants III-1763325, III-1909323, III-2106758, and SaTC-1930941. K.S. is supported by a Cisco Research Award. G.H.C. is supported by NSF CAREER award #2047981.

**Author contributions.** Conceptualization: K.L., Y.D., and Y.Z. Investigation and experiments: K.L., Y.D., Y.Z., X.D., X.H., R.Z., K.D., and C.C. Writing – original draft: K.L., Y.D., and Y.Z. Writing – review & editing: H.P., K.S., L.S., J.L., Z.H., and P.S.Y. Extensive reviewing & editing: G.H.C.

For any correspondence, please refer to Hao Peng.

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
