# A   Additional Details on BOND

## A.1   Additional Dataset Information

### A.1.1   Real Data

**Cora** [66] is a citation graph with nodes representing machine learning papers and edges representing papers' citation relationships. The node features are sparse bag-of-words (BoW) vectors extracted from the paper document, and their labels represent one of the seven classes.

**Amazon** [67] is a segment of the Amazon co-purchase graph [56], where nodes represent goods and edges indicate that two goods are frequently bought together. Notably, node features are BoW-encoded product reviews, and class labels are given by the product category.

**Flickr** [81] datasets originates from NUS-wide[12], a real-world web image database from the National University of Singapore. The SNAP website[1] collected Flickr data from four different sources including NUS-wide, and generated an undirected graph. A node in the graph represents one image uploaded to Flickr. If two images share some common properties (e.g., same geographic location, same gallery, comments by the same user, etc.), one edge is made between these two nodes. The node feature is composed of a 500-dimensional vector of the images provided by NUS-wide. Eighty-one tags of each image are manually merged into seven classes, and each image is assigned to one of the seven classes.

**Weibo** [86] is a user-posts-hashtag graph from Tencent-Weibo, a Twitter-like platform in China. This dataset collects information from 8,405 users with 61,964 hashtags. We use the user-user graph [2] provided by the author, which connects users who used the same hashtag. Temporal information was used to label the users. If a user made at least five suspicious events, he/she is labeled as a suspicious user; if no suspicious event was made, he/she is a benign user. There are a total of 868 suspicious users and 7,537 benign users. The suspicious users are regarded as outliers in the graph. Since the ground truth was generated using time information, the timestamps are not used to create raw user features. Therefore, the raw feature vector has two parts: (1) for each user, the one-hot vectors of his/her posts are summed where each one-hot vector represents the location where a post was made. Then the #dimension of the summed vector is reduced to 100 using SVD and (2) for each user, the #dimension of the BoW vectors extracted from post texts is reduced to 300. The final node feature is the concatenation of the location vector and the BoW vector. Note that Weibo is a directed graph; the remaining datasets used in our benchmark are undirected graphs.

**Reddit** [44, 73] is a user-subreddit graph extracted from a social media platform, Reddit [3]. This public dataset consists of one month of user posts on subreddits [4]. The 1,000 most active subreddits and the 10,000 most active users are extracted as subreddit nodes and user nodes, respectively. This results in 168,016 interactions. Each user has a binary label indicating whether it has been banned by the platform. We assume that the banned users are outliers compared to normal Reddit users. The text of each post is converted into a feature vector representing their LIWC categories [61] and the features of users and subreddits are the feature summation of the posts they have, respectively.

**Disney** [57] and **Books** [65] come from the Amazon co-purchase networks [48]. Disney is a co-purchase network of movies, where the attributes include prices, ratings, number of reviews, etc. The ground truth labels (i.e., whether it is an outlier) are manually labeled by high school students by majority vote. The second dataset, Books, is a co-purchase network of books on Amazon, which has similar attributes to the Disney dataset. The ground truth labels are derived from `amazonfail` tag information. More information about the datasets can be found on the project website [5].

**Enron** [65] is an email network dataset extracted from [41]. Each email is regarded as a node, and the messages between email addresses represent edges. The email addresses having sent spam messages are taken as outliers. Each node contains 20 attributes describing aggregated information about the average content length, the average number of recipients, or the time range between two emails.

---

[1] http://snap.stanford.edu/

[2] https://github.com/zhao-tong/Graph-Anomaly-Loss/tree/master/data/weibo_s

[3] https://www.reddit.com/

[4] http://files.pushshift.io/reddit/

[5] https://www.ipd.kit.edu/~muellere/consub/

**DGraph** [32] is a large-scale attributed graph with 3M nodes, 4M dynamic edges, and 1M ground-truth nodes. The nodes represent user accounts in a financial company providing personal loan services, and the edge between two nodes represents one account that has added another account as an emergency contact. For all the accounts with at least one borrowing record, the outliers are the accounts with overdue history, and the inliers are the accounts without overdue. Note there are also 2M accounts/nodes without any borrowing at all. The 17 node features are encoded from the user profile information like age and gender.

### A.1.2   Random Graph Generation Method

We leverage a random graph generation method used in [36] to create an arbitrary OND graph for benchmarking. Specifically, the implementation in PyG [1] is used with 2 classes, node_homophily_ratio=0.5, average_degree=5 and num_channels=64. We use the generated random graphs to benchmark algorithms' efficiency and scalability. We generate a random graph **Gen_Time** with num_nodes_per_class=500 (1000 in total) as the graph data to test the runtime. To benchmark the scalability, we generate multiple random graphs **Gen_100**, **Gen_500**, **Gen_1000**, **Gen_5000** and **Gen_10000** with num_nodes_per_class equal to 50, 250, 500, 2500 and 5000, respectively.

### A.1.3   Outlier Injection Details

For injecting structural outliers, we use $p = 0.2$. For contextual outliers, we set $q$ equal to $m$ in structural outlier injection. The other parameters used in outlier injection are shown in Table 6. Note that $m$ is set to be approximate twice the degree of the graph. For real datasets, we keep a similar outlier ratio (i.e., the number of outliers injected is approximately 5% of the total number of nodes; as a reminder, for structural outliers, we inject $m \times n$ outliers and for contextual outliers, we inject $o$ outliers; $o = m \times n$ in our setting). We keep a similar number of outliers for generated graph datasets of various sizes. The statistics of the generated graphs are shown in Table 7.

Table 6: Parameters used in synthetic outliers injection.

|  | Cora | Amazon | Flickr | Gen_Time | Gen_100 | Gen_500 | Gen_1000 | Gen_5000 | Gen_10000 |
|---|---|---|---|---|---|---|---|---|---|
| **Degree** | 4.1 | 37.5 | 10.6 | 5 | 5 | 5 | 5 | 5 | 5 |
| **n** | 70 | 350 | 2240 | 10 | 1 | 1 | 1 | 1 | 1 |
| **m** | 10 | 70 | 20 | 10 | 10 | 10 | 10 | 10 | 10 |

Table 7: Statistics of generated datasets in BOND.

| Dataset | #Nodes | #Edges | #Feat. | Degree | #Con. | #Strct. | #Outliers | Ratio |
|---|---|---|---|---|---|---|---|---|
| **Gen_Time** | 1,000 | 5,746 | 64 | 5.7 | 100 | 100 | 189 | 18.9% |
| **Gen_100** | 100 | 618 | 64 | 6.2 | 10 | 10 | 18 | 18.0% |
| **Gen_500** | 500 | 2,662 | 64 | 5.3 | 10 | 10 | 20 | 4.0% |
| **Gen_1000** | 1,000 | 4,936 | 64 | 4.9 | 10 | 10 | 20 | 2.0% |
| **Gen_5000** | 5,000 | 24,938 | 64 | 5.0 | 10 | 10 | 20 | 0.4% |
| **Gen_10000** | 10,000 | 49,614 | 64 | 5.0 | 10 | 10 | 20 | 0.2% |

### A.2   Description of algorithms in the benchmark

**LOF [5].** LOF is short for the Local Outlier Factor. LOF computes the degree of an object as abnormality, and the degree depends on how isolated the object is with respect to its surrounding neighborhood. Note that LOF only uses node attribute information, and the neighborhood is composed of $k$-nearest-neighbors.

**IF [52].** Isolation Forest (IF) is a classic tree ensemble method used in outlier detection. It builds an ensemble of base trees to isolate the data points and defines the decision boundary as the closeness of an individual instance to the root of the tree. It only uses node attributes of data.

**MLPAE [64].** The MLPAE is a vanilla autoencoder with multiple layer perceptron (MLP) as encoder and decoder. The encoder takes the node attribute as the input to learn its low-dimensional embedding

---

[1] `https://pytorch-geometric.readthedocs.io/en/latest/modules/datasets.html#torch_geometric.datasets.RandomPartitionGraphDataset`

and a decoder reconstructs the input node attribute from the node embedding. The outlier score of a node is the reconstruction error of the decoder.

**SCAN [75].** SCAN is a structural clustering algorithm to detect clusters, hub nodes, and outlier nodes in a graph. Since the structural outliers exhibit clustering patterns on graphs, we use SCAN to detect clusters and the nodes in detected clusters are regarded as structural outliers in the graph. SCAN only takes the graph structure as the input.

**Radar [50].** Radar is an anomaly detection framework for attributed graphs. It takes the graph structure and node attributes as the input. It detects outlier nodes whose behaviors are singularly different from the majority by characterizing the residuals of attribute information and its coherence with network information. The outlier score of a node is decided by the norm of its reconstruction residual.

**ANOMALOUS [60].** ANOMALOUS performs joint anomaly detection and attribute selection to detect node anomalies on attributed graphs based on the CUR decomposition and residual analysis. It takes the graph structure and node attribute as the input, and the outlier score of a node is decided by the norm of its reconstruction residual.

**GCNAE [39].** GCNAE is the autoencoder framework with GCNs [40] as the encoder and decoder. It takes the graph structure and node attributes as input. The encoder is used to learn a node's embedding by aggregating its neighbor information. The decoder reconstructs the node attribute by applying another GCN to node embeddings and graph structures. Similar to MLPAE, the outlier score of a node is the reconstruction error of the decoder.

**DOMINANT [16].** DOMINANT is one of the first works that leverage GCN and AE for outlier node detection. It uses a two-layer GCN as the encoder, a two-layer GCN decoder to reconstruct the node attribute, and a one-layer GCN and dot product as the structural decoder to reconstruct the graph adjacency matrix. The reconstruction errors of both decoders are combined as the outlier scores of the nodes.

**DONE [4].** DONE leverages a structural and an attribute AE to reconstruct the adjacency matrix and node attribute. The encoders and decoders are composed of MLPs. The node embeddings and outlier scores are optimized simultaneously with a unified loss function.

**AdONE [4].** AdONE is a variant of DONE, which uses an extra discriminator to discriminate the learned structure embedding and attribute embedding of a node. The adversarial training approach supposes to better align the two different embeddings in the latent space.

**AnomalyDAE [25].** AnomalyDAE also utilizes a structure AE and attribute AE to detect outlier nodes. The structure encoder of AnomalyDAE takes both the adjacency matrix and node attribute as input; the attribute decoder reconstructs the node attribute using both structure and attribute embeddings.

**GAAN [11].** GAAN is a GAN-based outlier node detection method. It employs an MLP-based generator to generate fake graphs and an MLP-based encoder to encode graph information. A discriminator is trained to recognize whether two connected nodes are from the real or fake graph. The outlier score is obtained by the node reconstruction error and real-node identification confidence.

**GUIDE [80].** GUIDE is similar to DONE and AdONE with two different AEs, but it pre-processes the structure information before feeding it into the structure encoder. Specifically, node motif degree is used to represent the node structure vector which could encode higher-order structure information.

**CONAD [76].** CONAD is one of the BOND methods that leverage graph augmentation and contrastive learning techniques. It imposes prior knowledge of outlier nodes via generating augmented graphs. After encoding the graphs using Siamese GNN encoders, the contrastive loss is used to optimize the encoder, and the outlier score of the node is obtained by two different decoders like DOMINANT.

### A.3 Description of Evaluation Metrics

**ROC-AUC (AUC).** AUC computes the Area Under the Receiver Operating Characteristic Curve (ROC-AUC) from predicted outlier scores. The ROC curve is created by plotting the true positive rate (TPR) against the false positive rate (FPR) at various threshold settings. In the BOND benchmark,

we regard the outlier nodes as the positive class and compute the AUC for it. AUC equals 1 means the model makes a perfect prediction, and AUC equals 0.5 means the model has no class-separation capability. AUC is better than accuracy when evaluating the outlier detection task since it is not sensitive to the imbalanced class distribution of the data.

**Average Precision (AP).** AP summarizes the precision-recall curve as the weighted mean of precisions achieved at each threshold, with the increase in recall from the previous threshold used as the weight. AP is a metric that balances the effects of recall and precision, and a higher AP indicates a lower false-positive rate (FPR) and false-negative rate (FNR). FPR and FNR have equal importance for most outlier detection applications as more misclassified normal samples could worsen legit users' experience.

**Recall@k.** The outliers are usually rare in contrast to enormous normal samples in the data, and the outliers are of the most interest to outlier detection practitioners. We propose to use Recall@k to measure how well the detectors rank outliers over the normal samples. We set $k$ as the number of ground truth outliers in each dataset. The Recall@k is computed by the number of true outliers among the top-k samples in the outlier ranking list divided by $k$. A higher Recall@k score indicates a better detection performance, and Recall@k equals 1 means the model perfectly ranks all outliers over the normal samples.

**Runtime.** Due to the coverage of both classical algorithms and neural network methods, we consistently measure the model runtime as the duration between the experiment starts and ends, mimicking the real-world applications without specific differentiation between CPU and GPU time.

**GPU Memory.** Notably, GPU memory is often the bottleneck of machine learning algorithms due to its limitation in extension. In BOND, we report the max active GPU memory for running an algorithm.

# B    Additional Experimental Settings and Details

**Environment.** The key libraries and their versions used in the experiment are as follows: Python=3.7, CUDA_version=11.1, torch=1.10, pytorch_geometric>=2.0.3, networkx=2.6.3, numpy=1.19.4, scipy=1.5.2, scikit-learn=0.22.1, pyod=1.0.1, pygod=0.3.0.

**Hardware configuration.** All the experiments were performed on a Linux server with a 3.50GHz Intel Core i5 CPU, 64GB RAM, and 1 NVIDIA GTX 1080 Ti GPU with 12GB memory.

**More model implementation details.** To include a large number of algorithms, we build Python Graph Outlier Detection (PyGOD)[1] [53], which provides more than 10 latest graph OD algorithms; all with unified APIs and optimizations. We tried our best to apply the same set of optimization techniques to each dataset. For Radar and ANOMALOUS, we use gradient descent instead closed-form optimization provided in official implementation due to fairness and efficiency concerns. For all deep algorithms, we implement sampling and minibatch training on large graphs (e.g., Flickr). See our library source code for more details. Meanwhile, we also include multiple non-graph baselines (LOF and IF) from our early work Python Outlier Detection (PyOD) [92].

**Hyperparameter grid**

The hyperparameter space is shown in Table 8. The candidates of hyperparameters are listed in square brackets. In each trial, a value is randomly chosen among candidates. The results (mean, std, max) are reported among 20 trials.

Due to the large graph size, full batch training on Flickr cannot fit in single GPU memory. Minibatch training and different batch size, sampling size, and the number of epochs are used on Flickr. Because of the complexity of real datasets, automated balancing by the standard deviation for weight alpha cannot balance well. Thus, three candidates are attempted. As Reddit has a lower feature dimension, we reduce hidden dimension values on Reddit.

**How we determine the optimal performance in runtime comparison.**

The optimal performance is determined by the ROC-AUC score. Taking the computational cost into account, we expect a reasonable score within as few training epochs as possible. Thus, when the

---

[1]PyGOD: `https://pygod.org/`

Table 8: Hyperparameters in different algorithms. The values in "[]" are candidates. We present these common hyperparameters shared by multiple algorithms on the top, and also specify some algorithm-specific hyperparameters at the bottom. Refer to PyGOD doc for more details.

| Algorithm | Hyperparameter | Cora | Amazon | Flickr | Weibo | Disney | Books | Enron | DGraph | Reddit | Gen |
|---|---|---|---|---|---|---|---|---|---|---|---|
| **Common** | *dropout* | [0, 0.1, 0.3] | | | | | | | | | |
| | *learning rate* | [0.1, 0.05, 0.01] | | | | | | | | | |
| | *weight decay* | 0.01 | | | | | | | | | |
| | *batch size* | full batch | | 64 | full batch | | | | 64 | full batch | |
| | *sampling* | all neigh. | | 3 | all neigh. | | | | 3 | all neigh. | |
| | *epoch* | 300 | | 2 | 300 | | | | 2 | 300 | |
| | *alpha* | auto | | | [0.8, 0.5, 0.2] | | | | | | auto |
| | *hid. dim.* | [32, 64, 128, 256] | | | | [8, 12, 16] | | | | [32, 48, 64] | |
| **SCAN** | *eps* | [0.3, 0.5, 0.8] | | | | | | | | | |
| | *mu* | [2, 5, 10] | | | | | | | | | |
| **AnomalyDAE** | *theta* | [10, 40, 90] | | | | | | | | | |
| | *eta* | [3, 5, 8] | | | | | | | | | |
| **GAAN** | *noise dim.* | [8, 16, 32] | | | | | | | | | |
| **GUIDE** | *struct. hid.* | [4, 5, 6] | | | | | | | | | |

score converges (i.e., the score increment of consequence epochs is less than 0.5%), we mark the current epoch as optimal.

# C Additional Experimental Results

## C.1 Additional Results on Real Dataset Detection Performance

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

Table 14: Recall@k (%) comparison among OD algorithms on three datasets injected with contextual and structural outliers, where we show *the avg perf. $\pm$ the STD of perf. (max perf.)* of each. The best algorithm by expectation is shown in **bold**, while the max performance per dataset is marked with underline. k is set as the number of each type of outliers in labels. OOM denotes out of memory with regard to GPU (_G) and CPU (_C).

| Algorithm | Cora | | Amazon | | Flickr | |
|---|---|---|---|---|---|---|
| | Contextual | Structural | Contextual | Structural | Contextual | Structural |
| LOF | 12.9±0.0 (12.9) | 5.7±0.0 (5.7) | 1.1±0.0 (1.1) | 3.7±0.0 (3.7) | 9.0±0.0 (9.0) | 2.2±0.0 (2.2) |
| IF | 13.1±3.3 (20.0) | 3.8±2.0 (8.6) | 4.4±3.5 (13.4) | 2.1±0.7 (3.1) | 13.8±1.9 (17.7) | 2.5±0.3 (3.0) |
| MLPAE | 15.7±0.0 (15.7) | 7.1±0.0 (7.1) | **55.1**±0.1 (55.1) | 2.3±0.0 (2.3) | **29.8**±0.1 (30.0) | 2.9±0.0 (2.9) |
| SCAN | 2.8±1.1 (4.3) | 25.9±19.9 (61.4) | 2.2±0.4 (2.9) | **25.1**±26.8 (70.3) | 2.8±0.2 (3.1) | 59.7±37.1 (96.4) |
| Radar | 1.4±0.0 (1.4) | 0.6±0.7 (1.4) | 15.1±6.5 (20.6) | 1.5±1.1 (2.3) | OOM_G | OOM_G |
| ANOMALOUS | 1.8±1.4 (4.3) | 1.2±2.4 (8.6) | 5.9±5.4 (19.1) | 0.6±1.4 (4.6) | OOM_G | OOM_G |
| GCNAE | 15.7±0.0 (15.7) | 7.1±0.0 (7.1) | **55.1**±0.1 (55.1) | 2.3±0.0 (2.3) | 21.6±12.7 (30.1) | 2.8±0.2 (3.0) |
| DOMINANT | 8.4±3.1 (10.0) | 17.2±5.2 (30.0) | 4.9±0.8 (6.0) | 10.1±1.9 (10.9) | 3.6±0.2 (4.0) | **70.6**±15.7 (76.0) |
| DONE | 8.6±3.9 (18.6) | **31.4**±26.2 (82.9) | 11.6±4.0 (21.1) | 23.7±19.9 (73.1) | 22.4±3.0 (25.8) | 5.1±1.0 (6.7) |
| AdONE | 11.4±3.1 (17.1) | 13.4±7.2 (35.7) | 18.1±7.4 (30.6) | 10.4±9.8 (30.9) | 17.7±3.2 (23.9) | 4.4±1.4 (7.7) |
| AnomalyDAE | 12.0±2.9 (17.1) | 16.0±4.2 (22.9) | 23.5±21.4 (53.4) | 8.8±2.9 (12.9) | 10.1±10.5 (30.2) | 3.6±1.7 (6.9) |
| GAAN | 15.9±0.4 (17.1) | 8.2±0.6 (8.6) | 54.9±0.6 (56.3) | 3.8±0.3 (4.6) | **29.8**±0.1 (30.1) | 2.9±0.0 (3.0) |
| GUIDE | **16.9**±0.6 (17.1) | 8.5±0.3 (8.6) | OOM_C | OOM_C | OOM_C | OOM_C |
| CONAD | 9.1±2.4 (10.0) | 17.2±3.9 (18.6) | 5.3±0.8 (6.0) | 10.2±1.2 (10.9) | 7.8±1.9 (10.1) | 8.3±1.4 (9.7) |

## C.3 Additional Results on Efficiency and Scalability Analysis

Table 15: Time consumption (s) comparison among OD algorithms on five different numbers of epochs. For non-iterative algorithms, i.e., LOF, IF, and SCAN, we report the total runtime.

| Algorithm | 10 | 100 | 200 | 300 | 400 |
|---|---|---|---|---|---|
| **LOF** | 0.10 | 0.10 | 0.10 | 0.10 | 0.10 |
| **IF** | 0.09 | 0.09 | 0.09 | 0.09 | 0.09 |
| **MLPAE** | 0.04 | 0.46 | 0.82 | 1.37 | 1.74 |
| **SCAN** | 0.02 | 0.02 | 0.02 | 0.02 | 0.02 |
| **Radar** | 0.02 | 0.10 | 0.19 | 0.34 | 0.36 |
| **ANOMALOUS** | 0.02 | 0.09 | 0.17 | 0.26 | 0.36 |
| **GCNAE** | 0.06 | 0.49 | 0.96 | 1.45 | 1.94 |
| **DOMINANT** | 0.08 | 0.70 | 1.41 | 2.10 | 2.79 |
| **DONE** | 0.08 | 0.77 | 1.53 | 2.30 | 3.08 |
| **AdONE** | 0.10 | 0.91 | 1.81 | 2.71 | 3.62 |
| **AnomalyDAE** | 0.43 | 0.64 | 1.28 | 1.92 | 2.55 |
| **GAAN** | 0.06 | 0.49 | 0.98 | 1.47 | 1.97 |
| **GUIDE** | 50.77 | 51.92 | 53.40 | 54.27 | 55.21 |
| **CONAD** | 0.11 | 1.04 | 2.07 | 3.07 | 4.10 |

Table 16: GPU memory consumption (MB) comparison among deep algorithms on five different graph sizes (number of nodes). Note that GPU memory measurement does not apply to algorithms like LOF, IF, and SCAN.

| Algorithm | 100 | 500 | 1000 | 5000 | 10000 |
|---|---|---|---|---|---|
| **MLPAE** | 0.66 | 2.10 | 3.90 | 19.41 | 38.52 |
| **GCNAE** | 0.92 | 3.40 | 6.38 | 31.18 | 62.11 |
| **GUIDE** | 0.96 | 3.46 | 6.47 | 31.45 | 62.60 |
| **Radar** | 0.49 | 9.32 | 36.73 | 871.32 | 3450.88 |
| **ANOMALOUS** | 0.44 | 5.03 | 20.50 | 482.47 | 2293.76 |
| **DOMINANT** | 1.09 | 7.58 | 27.15 | 591.74 | 2324.48 |
| **DONE** | 1.95 | 10.93 | 32.74 | 624.41 | 2385.92 |
| **AdONE** | 1.99 | 11.38 | 33.60 | 657.54 | 2447.36 |
| **AnomalyDAE** | 1.21 | 10.54 | 36.94 | 794.81 | 3112.96 |
| **GAAN** | 0.90 | 9.51 | 36.87 | 871.17 | 3450.88 |
| **CONAD** | 1.39 | 8.77 | 29.48 | 604.32 | 2344.96 |

# D  Long-term Maintenance and Development Plan

We commit to maintaining and developing BOND and PyGOD in the long run, as many of our open-source outlier detection works (e.g., PyOD [92], SUOD [90], and TODS [45]). More specifically, we will focus on improving on two aspects of graph OD tasks, namely datasets (Appx. D.1) and algorithms (Appx. D.2)

## D.1  Enriching Graph OD Datasets

We will keep monitoring the coming datasets suited for BOND tasks, and enrich our testbed with more datasets. There are three main approaches for this:

1. ***Directly including new graph OD datasets***. We will keep checking graph OD papers to include their newly introduced datasets.
2. ***Adapting graph datasets for graph OD tasks***. As we have discussed in future directions in §5, we could repurpose existing graph datasets for OD. For instance, given a graph dataset with multiple types of transactions for node classification, we could combine the rare classes together as anomalies, and the common transactions as the normal class. Most of the time, we could find some semantic meaning for the combined rare classes, e.g., fraud and mistakes. This adaptation process has been widely used in tabular OD [23] and has proven to be useful [6]. Specifically, Open Graph Benchmark (OGB) [30], and therapeutic data commons (TDC) [31] can serve as natural sources for building graph OD datasets, and we will start from these repositories.
3. ***Planting more types of synthesized outliers into plain graphs.*** Our experimental results and analysis suggest that the existing synthesizing approaches are too naive and not similar to most organic outliers. Our future plan includes: 1) adopting other outlier generation approaches from [75, 3]; 2) generating outliers using learning-based methods like GAN [68].

With more graph OD datasets (e.g., # datasets $>= 20$), we could conduct more in-depth (group-wise and pairwise) statistical analysis [14], which has not been possible in BOND works. We will keep updating the benchmark site[1] for newly added datasets.

## D.2  Emerging Graph OD Algorithms

Regarding graph OD algorithms, we will keep maintaining and improving PyGOD in multiple aspects:

1. ***Monitoring and adding outlier node methods to PyGOD*** for both benchmark and general usage.
2. ***Optimizing its accessibility and scalability*** with the latest development in graph learning [35], which may bring us new insights into outlier node detection's scalability.
3. ***Incorporating automated machine learning*** to enable intelligent model selection and hyperparameter tuning [59, 93], which may unlock some interesting perspectives of graph OD.
4. ***Extending the scope from static attribute outlier node detection to more graph tasks***, e.g., outlier detection in edges and sub-graphs. This will lead to other interesting aspects of graph OD.

**Robustness and Quality**. While building PyGOD, we follow the best practices of system design and software development. First, we leverage the continuous integration by *GitHub Actions*[2] to automate the testing process under various Python versions and operating systems. In addition to the scheduled daily test, both commits and pull requests trigger the unit testing. Notably, we enforce all code to have at least 90% coverage[3]. Following the `PEP8` standard, we enforce a consistent coding style and naming convention, which facilitates community collaboration and code readability.

In the long term, we envision PyGOD could keep evolving to support more comprehensive benchmarking, as well as other graph detection tasks and benchmarks.

---

[1]`https://github.com/pygod-team/pygod/tree/main/benchmark`
[2]Continuous integration by GitHub Actions: `https://github.com/pygod-team/pygod/actions`
[3]Code coverage by Coveralls: `https://coveralls.io/github/pygod-team/pygod`