# OpenReview forum: "BOND: Benchmarking Unsupervised Outlier Node Detection on Static Attributed Graphs"
_NeurIPS.cc/2022/Track/Datasets_and_Benchmarks — NeurIPS 2022 Datasets and Benchmarks _

### Official Review · Reviewer_Luyv · 2022-07-13
**Important Work that Needs Improvement**

**Rating:** 6
**Confidence:** 4

**Strengths:**

* It is important to have a unified benchmark for node outlier detection to facilitate fair comparison between methods and understand the progress in the field

* code is open sourced with good documentation

* a variety of methods are included in the benchmark and code for future research

* the paper includes interesting observations drawn from the benchmark such as synthetic outliers might not generalize to organic outliers and low-order structure information are sufficient for detecting structural outliers.

**Weaknesses:**

* **definition of structural outlier might be too narrow** (only densely connected nodes), I would suggest a more general definition. in static graphs, there exist natural dense hubs such as community leaders and they are not necessarily outliers. No experiments are conducted on more complex structural outliers. Due to this narraw definition, the injection of sturctural outlier is too easy to detect. Such synthetic outliers can be perfectly detected by simply counting the node degree in a single pass of all edges with 0 parameter. one suggested variation would be to add anomalous community nodes: form a community for nodes which doesn't initially have a community structure and have such group of nodes as anomalies. This is more similar to the example claimed in the paper "Structural outlier is prevalent in real-world graphs as members of organized fraud gangs usually interconnect with each other to carry out malicious activities" (see line 118)


* **lack of datasets.** There are currently 5 real world datasets and only 2 of them has true (non-synthetic) anomalies. In general, lack of labelled dataset is a huge challenge in anomaly detection over graphs. However, there are other available labeled datasets in the dynamic graph anomaly detection literature which can be converted to static graphs. For example, some well-known datasets are
  1. Enron email dataset
  2. DARPA dataset
  3. UCI Message
  4. Digg social network
see [Fast and Accurate Anomaly Detection in Dynamic Graphs with a Two-Pronged Approach (KDD 2019)]( https://arxiv.org/pdf/2011.13085.pdf) and [AddGraph: Anomaly Detection in Dynamic Graph Using Attention-based
Temporal GCN](https://www.ijcai.org/proceedings/2019/0614.pdf) As this is a dataset track paper, it would be important to include more datasets.

* **some contribution claims are weak** (see highlights in Introduction Section). The additional performance measures such as computational time and GPU usage are welcome to be compared between different methods however this is not novel concept and it is in general measured in anomaly detection literature. Minor note: future directions should be reserved in discussion or conclusion section only.



**Additional Feedback:**

I would consider raising my score accordingly if the authors can:
  * add additional synthetic outliers, improve the definition of the structural outliers.
  * add more datasets such as the ones I discussed in weakness section.

[update] The authors have addressed my concern for lack of datasets and added more synthetic structural anomaly definition.

**Clarity:**

The paper is well-written and easy to understand


**Correctness:**

datasets are constructed / collected in a sound way.

some contribution claims are not really novel and the synthetic structural anomaly definition & experiments can be improved (see weakness section).



**Documentation:**

Yes, data collection details are listed in the Appendix and also a detailed maintenance plan is discussed. I can access and the code through the provided URL and it is documented well. There is sufficient detail to support reproducibility.

**Ethics:**

There are no ethical concerns.

**Relation To Prior Work:**

Relation to prior work has been discussed well.

**Summary And Contributions:**


This work aims to address the lack of a standard and unified benchmark for node outlier detection. To this end,

* the authors proposed the first comprehensive unsupervised node outlier detection benchmark for graphs (UNOD) and examined fourteen existing outlier detection methods covering both non deep learning, deep learning and GNN based ones.

* An open sourced library called PyGOD is also included to reproduce many of these OD algorithms and the evaluation pipeline.

* The authors then classifier node outliers into structural and contextual outliers and methods have uneven performance across these two types of outliers.

* GPU memory consumption and runtime are also considered on top of current performance measures such as ROC-AUC.

* the authors also points out some future research directions.

---

> ### Author Response · Authors · 2022-08-24
> **Response to Reviewer Luyv (paper revision Aug 24th)**
>
> **Q1**. Definition of structural outlier might be too narrow.
>
> We agree with the reviewer’s thoughts on the limitation of the structural outlier injection method used in previous work. To make the synthetic structural outlier more realistic and close to our definition. We follow the reviewer’s suggestion to form community structures on the graph by randomly removing  20% of edges in a randomly created fully-connected graph and regard all nodes in the resulting dense subgraph as structural outliers. We rerun experiments on graphs with new structural outliers, and most of the UNOD methods have worse detection performance while their relative performance on the same graph is almost unchanged. The updated content can be found in Sections 3.2, 4.2.1, 4.2.2, and Appx. A.1.3, and Tables 3 and 5.
>
> **Q2**. Lack of datasets.
>
> We thank the reviewer for noting the lack of real-world datasets issue and suggesting candidate datasets. Since all three datasets are for edge anomalies in dynamic graphs, they apply to our node outlier detection on static graph problem settings. Nevertheless, we added four real-world graphs to the benchmark (see Table 1) and discussed other datasets we investigated while preparing the benchmark in Section 3.3.
>
> **Q3**. The additional performance measures such as computational time and GPU usage are welcome to be compared between different methods however this is not a novel concept, and it is in general measured in anomaly detection literature.
>
> We agree that time and memory consumption comparison are common in some ML fields, while their usages in graph OD tasks are limited. To the best of our knowledge, none of the previous graph OD works has benchmarked all baselines’ performance in terms of efficiency and scalability. We believe our benchmark could give a holistic view of the pros and cons of different methods’ efficiency and scalability. Moreover, as discussed in Section 4.2.3, we also find some insights on efficiency and scalability optimization through benchmarking.
>
> **Q4**. Future directions should be reserved in the discussion or conclusion section only.
>
> We thank the reviewer for suggestions on the content organization. We have simplified and combined the future directions in Sections 5.1 and 5.2 to leave more space for problem definition and experiment results analysis.

---

> > ### Comment · Reviewer_Luyv · 2022-08-25
> > **Response to Authors of Paper134**
> >
> > I thank the authors for addressing my main concerns. I believe the added datasets and synthetic outlier are great additions to the work. I have raised my score to reflect the positive improvement. I hope the authors will continue to add more datasets and maintain the benchmark.

---

### Official Review · Reviewer_3eC8 · 2022-07-22
**A solid benchmark for node-level outlier detection. Some clarifications are needed.**

**Rating:** 7
**Confidence:** 4
**Clarity:** The paper is well written and easy to…

**Strengths:**

1. It is a well-written paper that addresses an important need within the fair comparison of node OD.
2. The analysis for efficiency and scalability is important but has been ignored in previous literature. This work is a pioneer for these evaluations and I believe it is worthwhile for the fairness community.
3. The datasets and the documents for this benchmark are complete and accessible.
4. Beyond the benchmark contribution, the empirical analysis the authors provide is instructive and highlights the potential direction of node OD.


**Weaknesses:**

1. For structural outlier, it is too strict to construct the outlier subgraphs as cliques, which could make this kind of outlier too easy to be detected. Intuitively, humans can be aware of the outlier subgraph if it is obviously denser than the rest part.

2. The outlier number for each dataset is confusing. As mentioned in Appendix A.1.3, the outlier number for each outlier type is m×n. But in Table 1 of the dataset subsection, the outlier number equals n.


**Additional Feedback:**

The best results in the experiments in the Appendix should be highlighted.

**Correctness:**

Not all datasets support the conclusion in Section 4.2.1 about “there exists a trade-off between algorithm stability and potential”. This claim is strong. For instance, in Table 3, the best method on the Weibo dataset performs well for both stability and potential.

The evaluation of the benchmark is appropriate.

For dataset construction, there is a conflict about the outlier number, which needs further clarification.


**Documentation:**

Sufficient details and a complete introduction of the usage are provided. All the datasets and baseline models used in this benchmark are accessible.

**Ethics:**

No.

**Relation To Prior Work:**

The relation to existing work is clearly discussed in the paper.

**Summary And Contributions:**

The paper presents a benchmark for node outlier detection, which contains 2 outlier generators to construct the dataset.13 object detection methods are included in the benchmark for effectiveness and efficiency comparison.

---

> ### Author Response · Authors · 2022-08-24
> **Response to Reviewer 3eC8 (paper revision Aug 24th)**
>
> **Q1**. For structural outlier, it is too strict to construct the outlier subgraphs as cliques, which could make this kind of outlier too easy to be detected. Intuitively, humans can be aware of the outlier subgraph if it is obviously denser than the rest part.
>
> We thank a lot for the reviewer for pointing out this issue. We have changed the structural outlier injection method to inject dense subgraphs instead of fully-connected graphs. The dense subgraphs are created by randomly removing 20% of the edges in the fully-connected graph to make it less obvious. We rerun experiments on graphs with new structural outliers, and most of the UNOD methods have worse detection performance while their relative performance on the same graph is almost unchanged. The updated content can be found in Sections 3.2, 4.2.1, 4.2.2, Appx. A.1.3, and Tables 3 and 5.
>
> **Q2**.The outlier number for each dataset is confusing. As mentioned in Appendix A.1.3, the outlier number for each outlier type is m×n. But in Table 1 of the dataset subsection, the outlier number equals n.
>
> To unify the number of outliers injected by each type, we use m×n to represent the number of each type of outlier. In Table 1, #Outliers equals $m \times n$ for the first three datasets, and the detailed configuration can be found in Table 6 in Appx. A.1.3
>
> **Q3**. Not all datasets support the conclusion in Section 4.2.1 about “there exists a trade-off between algorithm stability and potential”.
>
> We sincerely thank you for mentioning this point. We carefully examine the claim with new experiment results, and revise the claim in 4.2.1. Specifically, we tone down to suggest there is likely a tradeoff for deep graph methods.
>
> **Q4**. The best results in the experiments in the Appendix should be highlighted.
>
> We thank the reviewer very much for noting this issue. We have fixed it in our updated manuscript. We also want to take the chance to sincerely thank your suggestions for making UNOD more accessible and usable :)

---

> > ### Comment · Reviewer_3eC8 · 2022-08-29
> > **Thank you for the response**
> >
> > My concerns have been addressed.

---

### Official Review · Reviewer_gHk2 · 2022-07-26
**This work benchmarks a diverse set of unsupervised node outlier detection methods on several datasets for synthetic and organic outliers. The baseline methods span from classical to graph neural network models. The datasets include injected synthetic or organic outliers.**

**Rating:** 7
**Confidence:** 5

**Strengths:**

-	The considered problem has attracted a lot of attention and is a very important problem with many usecases in different application domains.
-	Different range of methods from general outlier detection methods to graph neural network based methods is evaluated. Different performance metrics have been analyzed. In addition to efficiency, scalability of the methods based on the runtime and GPU memory of different methods is also evaluated.
-	Two different categories of outliers are identified and the power of different methods for detecting each type of outliers have been analyzed.


**Weaknesses:**

-	There are some grammar issues throughout the paper.
-	There are some proposed future research directions. Although discussing future direction is insightful, it should not be considered as a contribution of the paper.
-	Nine deep node outlier detection methods have been selected as baselines. However, there is no discussion on the intuition behind selecting these methods among all other available methods. This also applies to the non-existence of enough explanation on the selection criteria of graph-based methods.
-	There is no reference or discussion that could consolidate Definition 2 (Structural Outlier); why densely connected nodes are considered outliers?
-	The datasets are mostly small or medium size.
-	There is no discussion on the contextual vs. structural outliers on datasets with real outliers (i.e., Weibo & Reddit). Considering the lack of investigation of different types of outliers on datasets with real outlies, what is the motivation behind presenting the two kinds of outliers?
-	The caption of Table 3: … five real-world benchmark datasets…  this is misleading; actually, only two datasets contain actual outliers and the other three have injected outliers that are synthetic.
-	It has been shown that the performance on synthetic outliers does not generalize to organic outliers. It would be nice if more efficient methods for injecting or generating outliers were discussed.
-	Observing that the synthetic outliers are not good indicator of the actual outliers, what is the importance of evaluating the performance of synthetic outliers based on separating different types of outliers?
-	Figure 2: if the sequence of the methods in the legend follows the sequence of the bars, it could improve readability.
-	Section 5 has almost summarizes the previously observed results in related domains such as anomaly detection in other formats like tabular data.


**Additional Feedback:**

The impact of different kinds of outliers on datasets with organic outliers could be investigated. Better methods for injecting outliers to follow the behavior of real outliers could be considered and analyzed. The comparison between classical anomaly detection methods vs. graph-based or deep methods could be more elaborated to better incentivize the usage of one category of the methods (based on the dataset characteristics of types of outliers).

**Clarity:**

Totally, the paper is well-written, there are only some grammar issues (e.g., Section 4: research questions are incomplete sentences.)

**Correctness:**

-	The paper is a benchmarking paper. The evaluations make sense, however it could improve for better quality. Also, the results could be better investigated and explained.

**Documentation:**

The details and documentation are available as the supplementary materials.

**Ethics:**

No issue.

**Relation To Prior Work:**

The discussion on the previous work is clear.

**Summary And Contributions:**

This work provides a comprehensive unsupervised node outlier detection benchmark for graphs. In particular, it evaluates 14 methods, ranging from classical to graph neural network models, on 5 different datasets with synthetic or organic outliers.
The paper provides a definition of two different types of outliers (structural and contextual) and injects synthetic outliers into the datasets based on the proposed definitions.
The paper also compares the methods based on the efficiency and scalability of the algorithms by runtime and GPU memory usage on synthetic graphs at different scales.

---

> ### Author Response · Authors · 2022-08-24
> **Response to Reviewer gHk2 (paper revision Aug 24th) - Part 1**
>
> **Q1**. There are some grammar issues throughout the paper.
>
> We thank the reviewer for the grammar suggestions. We have fixed the incomplete sentences in research questions and proofread the updated manuscript more carefully to improve writing quality.
>
> **Q2**. There are some proposed future research directions. Although discussing future direction is insightful, it should not be considered as a contribution of the paper.
>
> We want to clarify that the original presentation of future research directions may not be optimal, where we combined both *the opportunity of graph OD design* and *future benchmark plans* together. In the revision, we streamline the *opportunities of graph OD algorithms* and merge them with the experiment result discussion in Section 4.2. Also, we move the future benchmark directions to Section 5 (right after the conclusion). We hope this new paper organization can reduce confusion in future directions.
>
> **Q3**. Nine deep node outlier detection methods have been selected as baselines. However, there is no discussion on the intuition behind selecting these methods among all other available methods. This also applies to the non-existence of enough explanation on the selection criteria of graph-based methods.
> We thank the reviewer for bringing this issue up. We have added the following explanations to Section 3.4 in the updated submission.
>
> ​​By adding non-graph OD algorithms (LOF, IF, MLPAE) into the benchmark, we can investigate the advantages and deficiencies of graph-based OD algorithms in detecting node outliers. Similarly, incorporating three non-deep structural OD methods (clustering-based SCAN, MF-based Radar, and ANOMALOUS) helps us understand deep OD methods' performance on structural outliers. We select a wide array of GNN-based UNOD methods, including the vanilla GCNAE; the classic DOMINANT; AnomalyDAE, an improved version of DOMINANT; GUIDE and CONAD, two state-of-the-art methods in this category with different data augmentation techniques. Besides GNN-based UNOD methods, two methods encoding graph information using other models (DONE/AdONE and GAAN) are also included.
>
> **Q4**. There is no reference or discussion that could consolidate Definition 2 (Structural Outlier); why densely connected nodes are considered outliers?
>
> Thanks for the suggestion! We have added the following descriptions and references in Section 3.2 to consolidate Definition 2: the structural outlier describes the outlier patterns in many real-world applications where members of organized fraud gangs usually collude and act together to carry out malicious activities and thus formulate dense subgraphs on the graph [1]. For instance, the coordinated bot accounts retweeted the same tweet will form a densely-connected co-retweet graph [2, 3].
>
> **Reference**
>
> [1]​​ L. Akoglu, H. Tong, and D. Koutra. Graph based anomaly detection and description: a survey.Data mining and knowledge discovery, 29(3):626–688, 2015.
>
> [2] B. Hooi, H. A. Song, A. Beutel, N. Shah, K. Shin, and C. Faloutsos. Fraudar: Bounding graph fraud in the face of camouflage. In KDD, 2016.
>
> [3] D. Pacheco, P.-M. Hui, C. Torres-Lugo, B. T. Truong, A. Flammini, and F. Menczer. Uncovering coordinated networks on social media: Methods and case studies. ICWSM, 21:455–466, 2021
>
> **Q5**. The datasets are mostly small or medium size.
>
> Agreed! To address the reviewer’s concern, we added four real-world graphs to the benchmark (see Table 1) and discussed other datasets we investigated while preparing the benchmark in Section 3.3.
>
> **Q6**. There is no discussion on the contextual vs. structural outliers on datasets with real outliers (i.e., Weibo & Reddit). Considering the lack of investigation of different types of outliers on datasets with real outliers, what is the motivation behind presenting the two kinds of outliers?
>
> We appreciate the reviewer's suggestion on discussing the organic outliers with proposed taxonomies. We have added two paragraphs in Section 4.2.1 to explain the success and failure of UNOD methods in detecting organic outliers by connecting organic outliers to our taxonomies.
>
> **Q7**. The caption of Table 3: … five real-world benchmark datasets…  this is misleading; actually, only two datasets contain actual outliers and the other three have injected outliers that are synthetic.
>
> We have added four more graph datasets with organic outliers and renamed the three graphs with injected outliers as synthetic graphs/datasets across the paper.
>
> **Q8**. It has been shown that the performance on synthetic outliers does not generalize to organic outliers. It would be nice if more efficient methods for injecting or generating outliers were discussed.
>
> We have noticed this problem and discussed it in Opportunity 3 in Section 4.2.1.

---

> > ### Author Response · Authors · 2022-08-24
> > **Response to Reviewer gHk2 (paper revision Aug 24th) - Part 2**
> >
> > **Q9**. Observing that the synthetic outliers are not good indicator of the actual outliers, what is the importance of evaluating the performance of synthetic outliers based on separating different types of outliers?
> >
> > We thank the reviewer for the question. In Section 4.2.2, we mention that practitioners may only focus on the high-value anomalies in their scenarios (e.g., fraudsters). Also, many previous methods detect outliers from these two aspects (e.g., DOMINANT has an attribute decoder and a structure decoder). Based on the evaluation of different types of outliers, we find that none of the detectors can balance two aspects well, which can be a potential opportunity for future works.
> >
> > **Q10**. Figure 2: if the sequence of the methods in the legend follows the sequence of the bars, it could improve readability.
> >
> > In Fig. 2 (left), we sort each method based on the time used when the number of epochs is 400. Similarly, in Fig. 2 (right), we sort the methods based on the GPU memory when the graph size equals 10000. If we sort by the bars' sequence, the methods' order may be less explicit.
> >
> > **Q11**. Section 5 has almost summarizes the previously observed results in related domains such as anomaly detection in other formats like tabular data.
> >
> > As mentioned in Q2, we now streamline the presentation of opportunities of graph OD algorithms and move them to Section 4.2. Given both graph and tabular OD concern with detecting rare samples in the data, many high-level observations are also shared. We believe this consistency with tabular data points out some great directions to adapt tabular approaches to graph OD.
> >
> > **Q12**. The comparison between classical anomaly detection methods vs. graph-based or deep methods could be more elaborated to better incentivize the usage of one category of the methods (based on the dataset characteristics of types of outliers).
> >
> > We appreciate the advice from the reviewer. We add one paragraph in line 254 to discuss the pros and cons of different categories of detectors and state that we should analyze the outlier patterns before selecting outlier detectors in Opportunity 1.

---

> > > ### Comment · Reviewer_gHk2 · 2022-08-26
> > > **Good improvement**
> > >
> > > The paper has been modified in several aspects and the authors have made good improvements to the paper to increase its quality.
> > > It is valuable that the authors added several organic datasets and modified/added new methods to their experiments. Also, the writing has improved.
> > > My concerns are mostly addressed. However, it is nice to have a solid maintenance plan for including more datasets and benchmarks in the paper.

---

### Official Review · Reviewer_Ct8Y · 2022-07-27
**Benchmarking graph anomaly detection**

**Rating:** 6
**Confidence:** 4
**Correctness:** Paper is correct and sound.

**Strengths:**

Paper is very well written and easy to read
The code, framework for datasets seems to be packaged nicely and is easy to use


**Weaknesses:**

Small number of real world datasets: only 3 real world dataset with anomalies are included which also do not show the same patterns with the datasets with injected anomalies. Given the evaluation of anomaly detection methods is hard, and results show inconsistency and depend on dataset/domain, it could have been more appropriate to have a larger set of real world datasets included, maybe even some effort into categorizing them. For example, datasets used in ref 57 for evaluation can be included and/or in Kumar, Srijan, et al. "Rev2: Fraudulent user prediction in rating platforms.".

There is little/no discussion/recommendations on inconsistencies between the performance metrics used, e.g. recall @k in appendix and AUC in the main paper don't agree, ranking of methods changes, and hence conclusions based on that.



**Additional Feedback:**

NA

**Clarity:**

Very well written and well organized, easy to read.

"Since there is no dedicated algorithm for the isolated structural outlier, we would not discuss it in our benchmark"> what does this mean? there are methods that find outliers in graphs, so it is not clear what this statement is suggesting.



**Documentation:**

Well documented GitHub page.

**Relation To Prior Work:**

Well positioned but still some prior work are missing, e.g. Fraudar, which is understandable given the number of works on this topic.

**Summary And Contributions:**

A framework is presented for evaluating unsupervised graph anomaly detection methods that flag anomalous nodes in a given attributed graph. Synthetic graphs with injected anomalies and real world graphs with known anomalies are used as datasets. Multiple conclusions are presented based on the performance of the methods on these datasets, given the type of the method (deep, graph, features), what signals it uses (graph, features, etc), as well as type of injected anomalies.  This could be a very valuable resource but there are weaknesses that need to be addressed to make it impactful.

---

> ### Author Response · Authors · 2022-08-24
> **Response to Reviewer Ct8Y (paper revision Aug 24th)**
>
> **Q1**.  Given the evaluation of anomaly detection methods is hard, and results show inconsistency and depend on dataset/domain, it could have been more appropriate to have a larger set of real-world datasets included, maybe even some effort into categorizing them.
>
> We thank the reviewer for noting the lack of real-world and large-scale datasets. To address the reviewer’s concern, we added four real-world graphs, including a large graph with more than 3.7 million nodes, to the benchmark and discussed other datasets we investigated while preparing the benchmark in Section 3.3.
>
> **Q2**. There is little/no discussion/recommendations on inconsistencies between the performance metrics used, e.g. recall @k in appendix and AUC in the main paper don't agree, ranking of methods changes, and hence conclusions based on that.
>
> Thanks for the suggestion. AUC evaluates the comprehensive performance of all samples, while recall@k only focuses on the data points with high outlier scores. Thus, the ranking results can be slightly different. For instance, in Cora, AnomalyDAE is optimal regarding average AUC, and also ranks 4th (above average) with respect to average recall@k. Also, DONE is the best in both max AUC and max recall@k. Most of our observations are based on the comprehensive evaluation (AUC), but we also estimate average precision and recall@k to provide more aspects of the effectiveness of the algorithms.
>
> **Q3**. Since there is no dedicated algorithm for the isolated structural outlier, we would not discuss it in our benchmark"> what does this mean? there are methods that find outliers in graphs, so it is not clear what this statement is suggesting.
>
> We appreciate the opportunity to clarify isolated structural outliers. The isolated structural outliers are nodes that have only a few edges connecting to the other communities/clusters in the graph. Since the isolated structural outlier is different from the structural outlier defined in our paper and there is no dedicated algorithm detecting the isolated structural outlier, we did not consider it in our taxonomy and benchmark. We have updated Section 3.2 to explain the above issue.
>
> **Q4**. Well positioned but still some prior work are missing, e.g. Fraudar, which is understandable given the number of works on this topic.
>
> We thank the reviewer for mentioning Fraudar---a representative method in node outlier detection. We have cited the Fraudar as Ref. [29] in Section 2.1 and consider it as one of the representative methods in density-based outlier detectors. In the UNOD benchmark, we have incorporated only one method named SCAN to represent density-based detectors. In future versions, we will incorporate Fraudar into our PyGOD package.

---

### Official Review · Reviewer_sgGb · 2022-07-27
**can be good work after massive improvement**

**Rating:** 6
**Confidence:** 5
**Clarity:** yes

**Strengths:**


 The paper is good in organization.

 I appreciate the code and datasets used are open


**Weaknesses:**

(1)	The problem is defined on the Static Attributed Graph (Problem Definition in section 3.1), while the title claims “Benchmarking Node Outlier Detection on Graphs”.

(2)	Given 14 detectors tested, only 8 of them [1,4,6,7,16,23,32,33] are originally designed for Attributed Graph. The real motivation for comparison to non-Attributed-Graph methods is missing.

(3)	Given 5 datasets, 2 of them contain organic outliers, while 3 of them contain outliers generated by the authors. However, since the datasets play a very vital role in experiments, the limitations of the method of generating outlier should be discussed. The data quality needs more guarantee, instead of merely using random injection. Otherwise, how can it be convincing that the dataset with injected outliers are trustworthy to the community.

(4)	From the experiment reports, there is no new findings to outlier detection domain listed in [1] published in 2017.

(5)    The authorship is very confusing.

**Additional Feedback:**

no

**Correctness:**

uncertain
(1) limitations of the datasets with generated outliers are missing, leading to uncertain risky to the conclusions drawn.
(2) all datasets are in small scale (<10w nodes)
(3) common parameter settings for different detectors are the same, not sure if this is fair or unfair actually. Why not try also the default settings of each detector?

**Documentation:**

yes

**Relation To Prior Work:**

yes

**Summary And Contributions:**

By defining the problem of “Unsupervised Node Outlier Detection on the Static Attributed Graph”, the authors conducted experiments with 14 detectors on 5 datasets.
The main contribution is the 14 detectors’ performance reports with taking below factors into account:
(1)	the choice of detectors from different years (2000, 2007, 2012, 2014, 2016, 2017, 2018, 2019, 2020, 2021, 2022);
(2)	the choice of outlier types including: injected outliers and organic outliers;
(3)	the evaluation metrics: accuracy, runtime, memory usage;

---

> ### Author Response · Authors · 2022-08-24
> **Response to Reviewer sgGb (paper revision Aug 24th) - Part 1**
>
> **Q1**. The problem is defined on the Static Attributed Graph (Problem Definition in section 3.1), while the title claims “Benchmarking Node Outlier Detection on Graphs”.
>
> Thanks for pointing this out! We will update our title to “Benchmarking Node Outlier Detection on Static Attributed Graphs” after the review period. For now, we have updated the abstract, introduction, future direction, and conclusion to better reflect the scope of the work.
>
> **Q2**. Given 14 detectors tested, only 8 of them [1,4,6,7,16,23,32,33] are originally designed for Attributed Graph. The real motivation for comparison to non-Attributed-Graph methods is missing.
>
> We thank the reviewer for bringing this up. Since the synthetic outlier types include the structural outlier, which is solely decided by structural information, we would like to include the non-attributed graph methods to test their performance on structural outliers. Moreover, comparing non-attributed graph methods with attributed-graph methods will demonstrate whether attributed-graph methods have better structural and contextual outliers detection performance than non-attributed graph detectors. We have updated Section 3.4 for the motivation of comparing non-attributed graphs and discuss the non-attributed graph outlier detectors in Section 4.2.2.
>
> **Q3**.  Given 5 datasets, 2 of them contain organic outliers, while 3 of them contain outliers generated by the authors. However, since the datasets play a very vital role in experiments, the limitations of the method of generating outlier should be discussed. The data quality needs more guarantee, instead of merely using random injection. Otherwise, how can it be convincing that the dataset with injected outliers are trustworthy to the community.
>
> We appreciate the reviewer’s insight on the limitation of the synthetic outliers in UNOD evaluation. Due to the lack of ground-truth data in graph outlier detection, using synthetic datasets has become a common practice for many previous works [1,2,3,4]. Since the objective of our work is to benchmark previous works under a unified setting, we adopted the most-used outlier injection method as an evaluation setting. In the revised version, we updated the structural outlier injection method to make it more realistic, and we added four more datasets with organic outliers to emphasize the importance of benchmarking on real-world data. Meanwhile, we admit the limitation and randomness of the outlier injection methods, discuss them in Section 3.3 and call for more standardized outlier synthetic methods in Section 5.
>
> **Reference**
>
> [1] K. Ding, J. Li, R. Bhanushali, and H. Liu. Deep anomaly detection on attributed networks. In Proceedings of the SDM, pages 594–602. SIAM, 2019.
>
> [2] H. Fan, F. Zhang, and Z. Li. Anomalydae: Dual autoencoder for anomaly detection on attributed networks. In Proceedings of the IEEE ICASSP, pages 5685–5689, 2020.
>
> [3] Z. Chen, B. Liu, M. Wang, P. Dai, J. Lv, and L. Bo. Generative adversarial attributed network anomaly detection. In Proceedings of the ACM CIKM, pages 1989–1992, 2020.
>
> [4] X. Yuan, N. Zhou, S. Yu, H. Huang, Z. Chen, and F. Xia. Higher-order structure based anomaly detection on attributed networks. In Proceedings of the IEEE Big Data Conference, pages 2691–2700, 2021.
>
> **Q4**. From the experiment reports, there is no new findings to outlier detection domain listed in [1] published in 2017.
>
> If we may kindly follow up on this, [1] in our original paper refers to the textbook *Outlier Analysis* written in Charu Aggarwal 2017, and the most relevant content is Chapter 12 (“Outlier detection in graphs and networks”), which is mainly for non-deep methods. Distinctly, our proposed future research directions highlight the importance of unsupervised model selection in graph OD supported by empirical observations and the new scalability issues poised for deep models when GPU memory is restricted. We have updated our future directions in Sections 4 and 5 to better connect with existing literature.
>
> **Q5**. Authorship explanation.
>
> Thanks for this clarification question. UNOD, as a collaborative work among multiple universities, takes in the expertise of each coauthor. We add an acknowledgment section to reflect each author's contribution to this work.
>
> **Q6**.  All datasets are in small scale (<10w nodes)
>
> We have added a recently-released large-scale fraud graph dataset DGraph [1] with more than 3 million nodes to benchmark UNOD performance on a large-scale graph.

---

> > ### Author Response · Authors · 2022-08-24
> > **Response to Reviewer sgGb (paper revision Aug 24th) - Part 2**
> >
> > **Q7**. Common parameter settings for different detectors are the same, not sure if this is fair or unfair actually. Why not try also the default settings of each detector?.
> >
> > Thanks for the question. As discussed in Section 4.1, hyperparameter tuning remains challenging and unclear for unsupervised outlier detection. Many existing works show that the optimal setting is highly data-dependent [2]. Thus, for each graph OD algorithm, we create a hyperparameter grid (which already captures their default hyperparameter setting). In this way, we could report the average and best performance of an OD algorithm, providing a more holistic view.
> >
> > **Reference**
> >
> > [1] Huang, X., Yang, Y., Wang, Y., Wang, C., Zhang, Z., Xu, J., & Chen, L. (2022). DGraph: A Large-Scale Financial Dataset for Graph Anomaly Detection.
> >
> > [2] Zhao, Y., Rossi, R. and Akoglu, L., 2021. Automatic unsupervised outlier model selection. Advances in Neural Information Processing Systems, 34, pp.4489-4502.
> >
> > We also want to take this opportunity to thank you for these detailed comments on paper clarification and improvement. These great thoughts have significantly improved UNOD :)

---

### Official Review · Reviewer_72w5 · 2022-07-29
**A unified benchmark for UNOD**

**Rating:** 7
**Confidence:** 4
**Clarity:** This paper is well written.

**Strengths:**

This paper has a clear motivation. The benchmark is well documented, and the paper is also well written and easily followed.


**Weaknesses:**

This research makes a novel attempt to measure the performance of various OD methods using taxonomy node outliers. This taxonomy, however, is too intuitive and unrelated to realistic datasets. Thus, despite the fact that this taxonomy can more accurately assess the effectiveness of current methods on synthetic datasets, the result is difficult to apply to real-world scenarios. I suggest that authors connect this part to real-world datasets, for example, by attempting to analyze current real-world datasets using this taxonomy or by connecting Table 4 and the real-world datasets parts of Table 3.

This benchmark places a strong emphasis on efficiency and scalability measurements. The largest dataset used in this study, however, is Flickr, which has just 89K nodes. This number has a gap with numerous real-world scenarios, which may restrict the applicability of efficiency and scalability analysis. I therefore strongly suggest the authors include more substantially large-scale datasets in their experiments, such as Amazon [1] (more than 4M edges), YelpChi [1] (more than 3M edges), Elliptic [2] (more than 203K nodes) and DGraph [3] (more than 3M nodes).

[1]  Yingtong Dou, Zhiwei Liu, Li Sun, Yutong Deng, Hao Peng, and Philip S Yu. Enhancing graph neural network-based fraud detectors against camouflaged fraudsters. In Proceedings of the 29th ACM International Conference on Information & Knowledge Management, pages 315–324, 2020.

[2] Mark Weber, Giacomo Domeniconi, Jie Chen, Daniel Karl I Weidele, Claudio Bellei, Tom Robinson, and Charles E Leiserson. Anti-money laundering in bitcoin: Experimenting with graph convolutional networks for financial forensics. arXiv preprint arXiv:1908.02591, 2019.

[3] Xuanwen Huang, Yang Yang, Yang Wang, Chunping Wang, Zhisheng Zhang, Jiarong Xu, and Lei Chen. DGraph: A Large-Scale Financial Dataset for Graph Anomaly Detection. Preprint, 2022.

**Additional Feedback:**

See Weaknesses.

**Correctness:**

The evaluation methods and experiment design is appropriate and performed correctly.

**Documentation:**

Yes

**Ethics:**

No.

**Relation To Prior Work:**

There are one or two papers missing, but they aren't crucial omissions.

**Summary And Contributions:**

This paper presents a comprehensive unsupervised node outlier detection benchmark called UNOD that aims to understand graph OD algorithms’ performance better. It examines fourteen OD methods on real-world and synthetic datasets and different types of outliers. Additionally, this paper creates Python Graph Outlier Detection (PyGOD), including more than ten latest graph OD algorithms.

---

> ### Author Response · Authors · 2022-08-24
> **Response to Reviewer 72w5 (paper revision Aug 24th)**
>
> **Q1**. I suggest that authors connect this part to real-world datasets, for example, by attempting to analyze current real-world datasets using this taxonomy or by connecting Table 4 and the real-world datasets parts of Table 3.
>
> We appreciate the reviewer’s suggestion on analyzing the organic outliers via taxonomies. We have added two paragraphs in Section 4.2.1 to explain the success and failure of UNOD methods in detecting organic outliers by connecting organic outliers to our taxonomies.
>
> **Q2**. I strongly suggest the authors include more substantially large-scale datasets in their experiments.
>
> We thank the reviewer for pointing out the lack of large-scale datasets problem. We investigated the datasets like YelpChi and Amazon mentioned by the reviewer. However, the performance of most algorithms is worse than a random guess on them, which might be due to the outliers’ strong dependence on supervised signals (our work only benchmarks unsupervised methods). To address the reviewer’s concern, we added four real-world graphs (including a large-scale data named DGraph with 3,700,550 nodes) to the benchmark and discussed other datasets we investigated while preparing the benchmark in Section 3.3.
>
> Again, thanks a lot for these useful comments to make UNOD a more complete work!

---

> > ### Comment · Reviewer_72w5 · 2022-08-28
> > **Significant update in experimental section**
> >
> > Thanks for expanding the experimental section to include more datasets. It indeed makes it more rigorous and naturally leads to a new observation. I've updated my score accordingly.

---

### Author Response · Authors · 2022-08-24
**Summary of Our Responses and Contributions (as of paper revision Aug 24th) and Long-term Plan**

We sincerely thank all the reviewers for their encouraging and insightful comments. We have carefully read through them and provided corresponding responses individually.

We upload the revised draft and the supplementary material, with changes highlighted in blue for reviewers’ attention. We also provide the main content+supplmentary version in the supplementary.

**The primary changes are summarized below**:

-  **More datasets with organic outliers**: we add Disney, Books, Enron, and a large-scale graph DGraph with 3.7 million nodes. Consequently, this revision is based on nine benchmark datasets (see Table 1 and Appx. A.1 for details; newly added datasets are highlighted in yellow). New experiment results are presented in Table 3 and Sections 4.2.1 and 4.2.3.

- **Improved taxonomy definition and outlier injection method**: we give a more detailed definition and discussion of two outlier types and add an edge drop with a probability of 20% in the injection of structural outliers to reduce the obviousness. New changes are presented in Sections 3.2, 4.2.1, 4.2.2, and Tables 3 and 5.

- **Additional experiment results analysis**: we analyze the reason behind the success and failure of some UNOD methods for detecting organic outliers by connecting them to our defined taxonomies. The new analysis is presented in Section 4.2.1.

- **Refined future research directions**: we reorganize and polish the suggested directions for graph OD researchers to make it more coherent to experiment results analysis. The new changes can be found in Sections 4.2.1, 4.2.2, 4.2.3, and 5.

**We also want to reiterate the major contributions of our paper as follows**:

**Novelty and Importance**: we conduct UNOD, the first comprehensive node-level OD benchmark on static attributed graphs. As *Reviewers 72w5, gHk2, 3eC8, and Luyv* affirm, this paper “has a clear motivation” and “the considered problem has attracted a lot of attention” and “is a very important problem with many use cases in different application domains.”

**Comprehensiveness**: in addition to analyzing the detection accuracy of 14 graph node OD algorithms, we also include efficiency and scalability measure, where *Reviewer gHk2, 3eC8, and Luyv* find the analysis of them “is important but has been ignored in previous literature” and “a pioneer work for these evaluations for the fairness community”. More importantly, we point out the insufficiency of existing graph OD algorithms’ design and evaluation, where it is unclear how to use them in real-world applications.

**Accessibility**: along with the paper, we release PyGOD, the most comprehensive graph OD library with easy access and complete documentation. Reviewer sgGb, Ct8Y, and Luyv appreciate the effort and mention its “good documentation” and “being packaged nicely and easy to use for future research.”

**Long-term plan**: We commit to maintaining and enriching PyGOD and UNOD in long run, as many of our open-source AD works (e.g., PyOD [1], SUOD [2], and TODS [3]). We will keep monitoring the coming datasets that are suited for graph OD tasks, and enrich the testbed of UNOD.


**Reference**


[1] Zhao, Yue, Zain Nasrullah, and Zheng Li. "PyOD: A Python Toolbox for Scalable Outlier Detection." JMLR, 2019.

[2] Zhao, Yue, et al. "SUOD: Accelerating large-scale unsupervised heterogeneous outlier detection." MLSys, 2021.

[3] Lai, Kwei-Herng, Daochen Zha, Junjie Xu, Yue Zhao, Guanchu Wang, and Xia Hu. "Revisiting time series outlier detection: Definitions and benchmarks." NeurIPS Benchmark and Datasets. 2021.


**Summary of newly added datasets**


|            | \#Nodes   | #Edges    | #Feat. | Degree | \#Outliers | Ratio |
| ---------- | --------- | --------- | ------ | ------ | ---------- | ----- |
| **Disney** | 124       | 335       | 28     | 2.7    | 6          | 4.8%  |
| **Books**  | 1,418     | 3,695     | 21     | 2.6    | 28         | 2.0%  |
| **Enron**  | 13,533    | 176,987   | 18     | 13.1   | 5          | 0.4‰  |
| **DGraph** | 3,700,550 | 4,300,999 | 17     | 1.2    | 15,509     | 0.4%  |

---

### Author Response · Authors · 2022-08-27
**Appreciate your acknowledgement and More on the Maintenance&Dev. Plan (Rev. as of Aug 27th)**

We appreciate all the initial comments and the acknowledgment of the revision and responses. They are encouraging and insightful! We take this valuable chance to further improve our work and provide a new revision (Aug 27th) with the following changes.

**Maintenance and Development Plan**. As some reviewers suggest, we want to elaborate on our long-term plan for the project. We add a new section (Appx. E) in the revision with two primary highlights for future maintenance and development of the UNOD benchmark and PyGOD library:

- **Enriching graph OD datasets**: (1) *directly including new graph OD datasets*: We will keep checking graph OD papers to include their newly introduced datasets (2) *adapting generic graph datasets for graph OD tasks*: we could repurpose existing graph (node classification) datasets for OD tasks, which have been widely used in tabular OD and (3) *planting more types of synthesized outliers into plain graphs* by more advanced generative models, e.g., learning-based GAN.
- **Including latest graph OD algorithms in PyGOD**: (1) *monitoring and adding newcoming UNOD methods to PyGOD* (2) *incorporating automated machine learning* for unsupervised graph OD model selection and hyperparameter tuning and (3) *extending the scope from static attribute node OD to more graph tasks*, e.g., outlier detection in edges and sub-graphs.

With more graph OD datasets (e.g., # datasets >= 20), we could conduct more in-depth (group-wise and pairwise) statistical analysis, which has not been possible in UNOD works. We will keep updating the benchmark site for newly added datasets and algorithms.
In the long term, we envision PyGOD could keep evolving to support more comprehensive UNOD benchmarking as well as other graph detection tasks and benchmarks.

**Proofreading and Improvement**. We appreciate the reviewers' feedback, and have made another round of paper polishing and improvement.

**We are happy to address any pending questions and concerns, and sincerely thank everyone for their time and insights into UNOD. This project will not be in its current shape without your guidance :)**

---

### Meta-Review · Area_Chair_SESi · 2022-09-15

**Recommendation:** Accept
**Confidence:** 4

**Metareview:**

This paper focuses on the task of anomaly or outlier detection in graphs. It provides a formulation of the notion of anomalies that enables synthetic injection of them into existing data, in addition to the consideration of organic anomalies. While reviewers raised certain concerns about the benchmarking procedures, as well as the used data (e.g., the use of synthetic anomalies vs. organic ones) in the initial version, most of these have been addressed during the rebuttal and discussion period, and at this point all reviewers ultimately recommend acceptance. Therefore, I also recommend acceptance of the paper.

---

### Decision · Program_Chairs · 2022-09-16

Accept